

# Significant additional Antarctic warming in atmospheric bias-corrected ARPEGE projections

Julien Beaumet[1], Michel Déqué[2], Gerhard Krinner[1], Cécile Agosta[3], Antoinette Alias[2], and Vincent Favier[1]

[1]Univ. Grenoble Alpes, CNRS, Institut des Géosciences de l'Environnement, F-38000, Grenoble, France
[2]CNRM, Université de Toulouse, Météo-France, CNRS, Toulouse, France
[3]Laboratoire des Sciences du Climat et de l'Environnement, LSCE-IPSL, CEA-CNRS-UVSQ, Université Paris-Saclay, F-91190 Gif-sur-Yvette, France

*Correspondence to:* Julien Beaumet (Julien.Beaumet@univ-grenoble-alpes.fr)

**Abstract.** In this study, we use run-time bias-correction to correct for ARPEGE atmospheric model systematic errors on large-scale atmospheric circulation. The bias correction terms are built using the climatological mean of the adjustment terms on tendency errors in an ARPEGE simulation relaxed towards ERA-Interim reanalyses. The improvements with respect to the AMIP-style uncorrected control run for the general atmospheric circulation in the Southern Hemisphere are significant

for mean state and daily variability. Comparisons for the Antarctic Ice Sheet with the polar-oriented regional atmospheric models MAR and RACMO2 and *in-situ* observations also suggest substantial bias reduction for near-surface temperature and precipitation in coastal areas. Applying the method to climate projections for the late 21$^{st}$ century (2071-2100) leads to large differences in the projected changes of the atmospheric circulation in the Southern high latitudes and of the Antarctic surface climate. The projected poleward shift and strengthening of the southern westerly winds are greatly reduced. These changes

result in a significant 0.7 to 0.9 K additional warming and a 6 to 9% additional increase in precipitation over the grounded ice sheet. The sensitivity of precipitation increase to temperature increase (+7.7 and +9%.K$^{-1}$) found is also higher than previous estimates. Highest additional warming rates are found over East Antarctica in summer. In winter, there is a dipole of weaker warming and weaker precipitation increase over West Antarctica, contrasted by a stronger warming and a concomitant stronger precipitation increase from Victoria to Adélie Land, associated with a weaker intensification of the Amundsen Sea Low.

## 1 Introduction

The Antarctic Ice Sheet (AIS) contribution to sea-level rise (SLR) has increased dramatically since the 1990s (Velicogna, 2009; Shepherd et al., 2018). The largely positive AIS surface mass balance (SMB), for which positive but insignificant trends have been reported during the second part of the 20$^{th}$ century (Lenaerts et al., 2016; Palerme et al., 2017; King and Watson,

2020), now fails to compensate for increasing ice losses of the West Antarctic Ice Sheet (WAIS) (Velicogna, 2009; Pritchard





et al., 2012; Shepherd et al., 2018). During the course of the 21$^{st}$ century, the AIS contribution to SLR is expected to increase (Ritz et al., 2015), possibly dramatically (Pollard et al., 2015). Regarding AIS SMB, there is currently a good agreement on the expectation that it will increase at a rate of $5\pm1$ % K$^{-1}$ (Agosta et al., 2013; Ligtenberg et al., 2013; Frieler et al., 2015; Krinner et al., 2014), mainly as a result of increase in snowfall linked to higher water vapour saturation pressure in a warmer climate.

In this regard, it is crucial to reduce the uncertainties about the evolution of the future Antarctic climate, and particularly SMB, in order to assess its negative contribution to SLR and to better constrain surface forcings for ice dynamics, and ocean and ice-shelves interactions studies.

Due to a lack of *in-situ* measurements and the difficulty to measure SMB from space (Favier et al., 2013; Thomas et al., 2017), polar-oriented regional climate models (RCM) driven by climate reanalyses are deemed the most reliable, and are the

most commonly used method to provide estimates of the current Antarctic SMB (e.g., Agosta et al., 2019; van Wessem et al., 2018). For future Antarctic SMB projections, RCMs are usually driven by output from coarser resolution coupled Atmosphere-Ocean General Circulation Model (AOGCMs), such as those involved in the Coupled Model Intercomparison Project, phase 5 (CMIP5, Taylor et al., 2012) or the ongoing phase 6 (CMIP6, Eyring et al., 2016). Since all these models show substantial biases (Gleckler et al., 2008; Flato et al., 2013), there are large uncertainties associated with the dynamical downscaling using

RCMs of their future projections. This is particularly relevant for Antarctica, firstly because many state-of-the-art AOGCMs fail to reproduce Southern Hemisphere Sea-Ice Extent (SIE) seasonal cycle and recent trends (Turner et al., 2013; Mahlstein et al., 2013). This is concerning as Sea Surface Conditions (SSC) around Antarctica were shown to have larger instantaneous control on future Antarctic climate than increases in Greenhouse Gases (GHG) concentration (Krinner et al., 2014). Secondly, because in the Southern mid-latitudes, the CMIP5 ensemble mean shows a large ($\sim$3°) equatorward bias on the position of

the surface westerly winds maximum (or "jet") (Bracegirdle et al., 2013) and there are large uncertainties associated with the atmospheric circulation change signals suggested by these models, as there is an historical state dependence: models with a larger equatorward bias show a larger poleward shift in a warmer climate (Bracegirdle et al., 2013). For Antarctic climate change assessment, it is of prime importance to evaluate the poleward shift of the westerlies and variations of their intensity, which are among the most salient expected consequences of GHG forcing for the Southern Hemisphere atmospheric circulation

(Arblaster and Meehl, 2006; Miller et al., 2006; Fyfe and Saenko, 2006). The associated storm track changes have influenced regional warming (Nicolas and Bromwich, 2014) and SMB changes (Medley and Thomas, 2019) in Antarctica and other southern high latitude regions (e.g., Verfaillie et al., 2019), and will continue to do so (Bracegirdle et al., 2013). Besides, wind-driven oceanic currents in the Amundsen Sea sector influence the rate of ice shelf basal melt (Rignot et al., 2013), which can enhance ice discharge in this sector (Pritchard et al., 2012; Fürst et al., 2016) where large ice losses have been reported recently

(e.g., Shepherd et al., 2018).

Another possibility for the downscaling of climate model outputs is the use of variable-resolution or stretched-grid atmosphere-only GCMs (VarGCM : e.g., Fox-Rabinovitz et al., 2006; McGregor, 2015). For projections, the anomaly method, which consists in driving the atmospheric model with observed SSC for historical climate and bias-corrected SSC coming from AOGCM scenarios for future projections, has been extensively used with such models (e.g., Gibelin and Déqué, 2003; Déqué, 2007;

Krinner et al., 2008; Beaumet et al., 2019a). Using these methods, the uncertainties on base-line historical climate as well as





on climate change signals are reduced. However, even when driven by observed SSC, biases in atmospheric models remain substantial. For instance, the CMIP5 AMIP (Atmospheric Model Intercomparison Project, Gates, 1992) ensemble mean still shows the classical double Intertropical Convergence Zone (ITCZ) problem, even if it is reduced with respect to the CMIP5 coupled model ensemble (Li and Xie, 2012).

Because all regional and global climate model bear some biases, and provide information at horizontal resolution too coarse for impact studies, outputs from climate scenarios are generally bias-corrected (or bias-adjusted) and downscaled *a posteriori* using statistical methods (Hall, 2014; Maraun and Widmann, 2018). However, such methods fail to correct for biases associated with atmospheric general circulation errors (Eden et al., 2012; Stocker et al., 2015; Maraun et al., 2017) or for biases due to poorly represented feedback processes in a warming climate (Maraun et al., 2017). Bias stationarity is a strong hypothesis

needed to justify bias-correction of model errors on atmospheric general circulation. Due to the lack of such justifications, empirical run-time bias correction of systematic errors on atmospheric general circulation using the statistics of a nudged simulation towards climate reanalysis such as done in Guldberg et al. (2005) or in Kharin and Scinocca (2012) has been restricted so far to seasonal forecasting applications. Krinner and Flanner (2018) have shown that state-of-the-art coupled AOGCMs show striking stationarity of large-scale atmospheric circulation biases under strong warming scenarios. Recently,

(Krinner et al., 2019b) demonstrated the added-value of run-time bias correction for future projections using perfect model tests. Taking advantage of these findings, we here apply run-time bias correction using the statistics on tendency errors of atmospheric variables in an ARPEGE (Déqué et al., 1994) simulation nudged towards ERA-Interim reanalysis (Dee et al., 2011), following closely the method described in Guldberg et al. (2005). The method is presented in section 2. In section 3.1, we present the improvement obtained for the representation of the Southern Hemispheric atmospheric general circulation as

well as for Antarctic surface climate and SMB. In section 3.2, we present the climate change obtained for the late 21[st] century and compare them with climate changes obtained in the control simulations performed without atmospheric bias-correction. The bias reduction obtained for historical climate and the difference in Antarctic projected climate change for late 21[st] century are discussed in section 4.

## 2    Data and Methods

### 2.1    CNRM-ARPEGE set-up

In this study, the configuration of the CNRM-ARPEGE model is the same as in Beaumet et al. (2019a). The 6.2.4 version of ARPEGE (Déqué et al., 1994), a spectral primitive equation AGCM, is used with a T255 truncation, a streching pole on the centre of the East Antarctic Plateau (80°S, 90°E) and a 2.5 stretching factor, which means that the horizontal resolution is increased by a factor 2.5 with respect to a regular grid of equal size at the stretching pole. With this setting, the horizontal

resolution over Antarctica varies between 30 km at the stretching pole and 45 km over the northernmost parts on the Antarctic Peninsula (AP). The horizontal resolution at the antipodes in the Northern Hemisphere is 135 km. The atmosphere is discretized into 91 sigma-pressure vertical levels. The surface processes are solved by SURFEX-ISBA-ES surface scheme (Noilhan and Mahfouf, 1996). Over snow-covered surfaces, a 3-layer intermediate complexity snow scheme (Boone and Etchevers, 2001)





is used, which explicitly accounts for the evolution of the surface albedo, heat transfer through the snow layers and refreezing of liquid water. Over the ocean, a 1D version (that is, without sea-ice advection) of the sea-ice model GELATO (Mélia, 2002) is used. In each simulation, a spin-up phase of two years for the atmosphere is considered and these two years are dismissed for the analysis. ARPEGE set-up and prescribed oceanic surface conditions used in this study are identical to those used in

Beaumet et al. (2019a).

## 2.2    Empirical bias correction of the AGCM

Following the method presented by Guldberg et al. (2005), we use the climatological mean correction terms of a nudged ARPEGE simulation to build a climatology of the tendency errors of the atmospheric model in a first step. A recent study applying a similar method has been performed for Antarctic climate change (Krinner et al., 2019a) using the LMDZ model,

the atmospheric component of the IPSL coupled climate model. The second step consists in adding a term derived from the climatology of the model drift to the prognostic equations of the model, in order to correct in-line (at each time step) selected atmospheric state variables (see below). More precisely, in the first nudged experience, ARPEGE was relaxed (Jeuken et al., 1996) towards ERA-Interim reanalysis (Dee et al., 2011) over 18 years (1993-2010) in order to use the most reliable period of the reanalysis over the Southern Hemisphere. In this simulation, initial first guest for a given prognostic variables $\psi$ are relaxed

towards the reanalysis reference data following Guldberg et al. (2005) :

$$\psi(t) = \psi^{\star}(t) + \Delta t \frac{\psi^{REF}(t) - \psi^{\star}(t)}{\tau}. \tag{1}$$

The upper index $\star$ stands for the prognostic solution of the atmospheric model dynamics and physics for the corresponding time step, while $REF$ indicates the reference variable, from ERA-Interim reanalysis in this case, towards which the model is nudged. The relaxation time ($\tau$) for the nudging is 72 hours for each variable, with an update every 6 hours using a linear

interpolation. The nudged variables are the following : air temperature, air specific humidity, logarithm of surface pressure, divergence and vorticity. The term $\left[\psi^{REF}(t) - \psi^{\star}(t)\right]/\tau$ in (1) is the estimate of the tendency residual and is stored in memory at each time step in order to build the correction terms :

$$G = \left[\overline{\frac{\psi^{REF}(t) - \psi^{\star}(t)}{\tau}}\right]^{AC} \tag{2}$$

In (2), the $AC$ exponent indicates that climatological mean is applied to the tendency residuals in order to produce a seasonally

and spatially varying correction term which can be seen as a climatology of the free atmospheric model bias with respect to the reference data set. In a second experiment, this correction term $G$ is then added at each time step to the atmospheric model prognostic equations for the variables mentioned above :

$$\psi^{C}(t) = \psi^{\star}(t) + G \tag{3}$$

which yields the empirically bias-corrected solution $\psi^{C}$. The 72 h value for $\tau$ in the first nudged experiment was chosen after a

few sensitivity tests, as this value yields acceptably small errors in the corrected experiment and weak bias correction terms of the order of a few percent of typical physical tendencies. Moreover, Guldberg et al. (2005) found that small values of $\tau$ (e.g., 6



hours) are not recommended for variables that are poorly constrained in the climate reanalyses, such as the divergence. In the bias-corrected experiment, variables are corrected only above the planetary boundary layer (around 1500 m above sea level), with a progressive transition to uncorrected variables towards the lowest layers (around 100 m).

## 2.3 Simulation setup

5   In this work, we use a set of 6 simulations, summarized in Table 1. Three simulations use atmospheric bias correction, whereas the three other simulations are the uncorrected reference. Both of these subsets consist of one AMIP-type present-day simulation using observed sea-surface conditions and atmospheric boundary conditions (greenhouse gas concentrations etc.), and of two simulations for the 2071-2100 period under the RCP8.5 forcing scenario, using oceanic surface anomalies from coupled CMIP5 projections.

10   Observed Sea Surface Temperatures (SST) are used for the AMIP-type control run (denoted ARP-AMIP in the following) and for the present-day run with atmospheric bias correction (denoted ARP-AMIP-AC). As already mentioned above, the 1D version of sea-ice model GELATO is used over the sea-ice area. However the sea-ice concentration (SIC) is nudged towards observed or bias corrected (for future projections) SIC. In order to obtain a consistent sea-ice thickness (SIT) with concentration, especially between recent climate simulations and projected climate, the SIT is prescribed using a simple parametrization 15  (Krinner et al., 1997, 2010) presented in Beaumet et al. (2019b). SST and SIC in future projections are bias-corrected following methods and recommendations from Beaumet et al. (2019b).

  For the future projections, prescribed SST and SIC changes come from MIROC-ESM and NorESM1-M model under their radiative concentration pathways RCP8.5 (Moss et al., 2010). The reason for this choice and a more complete analysis are presented in Beaumet et al. (2019a). To summarize, these two models were chosen among the CMIP5 ensemble because they 20  display very different changes in winter Sea-Ice Extent (SIE) in their RCP8.5 projection for the late 21[st] century (2071-2010) around Antarctica (respectively -45 % and -14 %). Since we bias-correct SSC from the AOGCM scenarios, our choice of model is guided by the desire to cover a large range of possible future evolution of SSC around Antarctica rather than by their skills for SSC in present climate. The use of bias-corrected SSC in our future projections is justified by the need to remain consistent with the bias correction terms for the atmospheric model derived in the present climate with an experiment using observed SSC. 25  We combine corrected SSC from the two chosen AOGCMs and bias-corrected or uncorrected atmospheric model to produce four different future climate projections, which are presented and compared in this study. The suffix "OC" in the simulation names (see Table 1) indicates that only the oceanic boundary conditions are bias-corrected, whereas the suffix "AOC" indicates that in addition to using bias-corrected oceanic boundary conditions, the atmospheric correction is also applied.

## 2.4 Evaluation method for historical climate

30   In section 3.1, we present the improvement in ARP-AMIP-AC with respect to ARP-AMIP for the representation of the atmospheric general circulation in the Southern Hemisphere (south of 20°S) and for the Antarctic surface climate over the 1981-2010 period. For atmospheric general circulation, both simulations are compared to the ERA-Interim reanalyses (ERA-I in the following, Dee et al., 2011), with evaluation of temperature, geopotential heights and specific humidity at different pressure levels





**Table 1.** Summary of the period, sea surface conditions, greenhouse gazes concentration and reference historical simulation for each ARPEGE future projections and historical simulation presented in this study. AC, OC, and AOC acronyms stand for Atmospheric and/or Oceanic Correction. * ARP-AMIP, ARP-MIR-21-OC and ARP-NOR-21-OC are also described in Beaumet et al. (2019a)

| Simulations | Period | Atm. | SSC | GES | Reference for hist. climate |
|---|---|---|---|---|---|
| ARP-AMIP | 1981-2010 | Uncorr. | Observed | historical | - |
| ARP-AMIP-AC | 1981-2010 | Bias-corr. | Observed | historical | - |
| ARP-NOR-21-OC | 2071-2100 | Uncorr. | Bias-corr. NorESM1-M RCP8.5 | RCP8.5 | ARP-AMIP |
| ARP-MIR-21-OC | 2071-2100 | Uncorr. | Bias-corr. MIROC-ESM RCP8.5 | RCP8.5 | ARP-AMIP |
| * ARP-NOR-21-AOC | 2071-2100 | Bias-corr. | Bias-corr. NorESM1-M RCP8.5 | RCP8.5 | ARP-AMIP-AC |
| * ARP-MIR-21-AOC | 2071-2100 | Bias-corr. | Bias-corr. MIROC-ESM RCP8.5 | RCP8.5 | ARP-AMIP-AC |

for the representation of the mean state. The relative root mean square error (RMSE, denoted $E$ in the following) reduction obtained for the new ARP-AMIP-AC simulation with respect to the previous uncorrected reference is calculated using :

$$\Delta_r E = 1 - \frac{E_{\mathrm{ARP-AMIP-AC}}}{E_{\mathrm{ARP-AMIP}}} \qquad (4)$$

Besides, an assessment of the high frequency variability in ARP-AMIP-AC, ARP-AMIP and ERA-I has been performed using an artificial neural network also called self-organizing map (SOM, Kohonen, 1990, 2013), as already done in other climate studies (e.g., Reusch et al., 2007; Sheridan and Lee, 2011; Krinner et al., 2014). The unsupervised machine-learning algorithm was given as input the daily sea-level pressure (SLP) maps of the first 10 years (1981-1990) of each ARPEGE simulation pre-

10 sented in this study and from the corresponding period in ERA-I. The 20 typical circulation patterns (also called Best Matching Units, BMU) that were identified after this first step are presented on a 5x4 hexagonal grid in Fig. 2. In a second step, each daily SLP map from each simulation is attributed to the closest BMU using the same distance metric as used to determine the BMU in the first step.

For surface climate, ERA-I reanalyses were shown to have substantial biases in Antarctic near-surface atmospheric tempera-

15 tures ($T_{2m}$ hereafter), especially over the East Antarctic Plateau (Fréville et al., 2014; Dutra et al., 2015). On the other hand, Antarctic $T_{2m}$ and SMB in polar-oriented RCMs such as MAR and RACMO2 (Van Meijgaard et al., 2008) have been success-fully validated against *in situ* observations in many studies (Van Wessem et al., 2014; Agosta et al., 2019; van Wessem et al., 2018) and were found to generally outperform climate reanalyses for Antarctic precipitation and SMB. Therefore, ARPEGE $T_{2m}$, precipitation and SMB are evaluated against MAR (Agosta et al., 2019) and RACMO2 (van Wessem et al., 2018) ERA-I

driven simulations and *in situ* data (for temperature only) from the SCAR READER data base (Turner et al., 2004), as in Beaumet et al. (2019a). Significance of the differences is assessed using double-sided t-tests as in Beaumet et al. (2019a).





# 3  Results

## 3.1  Evaluation for Present Climate

In this section, the results of the evaluation for the representation of atmospheric general circulation mean state and high frequency variability are first presented and are followed by the results of the evaluation for surface climate.

### 3.1.1  Atmospheric General Circulation : mean state

The errors with respect to winter and summer ERA-I SLP for ARP-AMIP and ARP-AMIP-AC over the 1981-2010 period can be seen in Fig. 1. As already presented in Beaumet et al. (2019a), the uncorrected ARP-AMIP control run is low biased in the Southern mid-latitudes, especially in the Pacific sector, while it underestimates the depth of the circum-antarctic troughs (that is, there is a positive pressure bias), particularly the Amundsen Sea Low (ASL). In ARP-AMIP-AC, most of these errors are removed and only a slight positive SLP bias around the Antarctic coasts remains in winter (JJA, Fig. 1a). The magnitude of the bias is substantially reduced in the Amundsen Sea sector. The slight remaining positive bias in winter around the Antarctic coasts for ARP-AMIP-AC is also found in the 850 and 500 hPa geopotentials height (not shown).

We evaluate the improvement in ARPEGE skills associated with run-time bias correction trough the RMSE reduction ($\Delta_r E$) south of 20°S with respect to ERA-I over the whole period 1981-2010 (Table 2). In order to assess the method independently from the period from which the correction terms where derived (1993-2010), we also assessed the $\Delta_r E$ for a few key variables in ARP-AMIP-AC for the earliest and independent 1981-1992 period (Table A1). The results of the evaluation using any of the two periods are similar and large RMSE reduction are found. For SLP, the RMSE reduction in ARP-AMIP-AC with respect to ARP-AMIP ranges between 50 and 90% with the lowest improvement in winter. The largest improvements are found for mid and upper tropospheric temperatures with RMSE decrease around 90% in all seasons. At 200 hPa, a large cold bias, increasing with height, was found in the southern Tropics and mid-latitudes in the ARP-AMIP simulation. In ARP-AMIP-AC, these biases are completely suppressed. However, a larger warm bias at 200 hPa ($\sim$ 2K) is present over Antarctica in winter and spring (Fig. A1). RMSE scores indicate little bias reductions for 850 hPa temperatures (T850, Fig. A2), 850 and 500 hPa specific humidity (Q850 and Q500, not shown). Below 850 hPa, the atmospheric state is progressively uncorrected in ARP-AMIP-AC.

The position and the value of the maximum of 850 hPa zonal wind component, referred to as westerlies maximum position (WMPOS) and strength (WMSTR) henceforth are presented in Table 3. In the uncorrected ARP-AMIP simulation, the westerlies maximum is characterized by significant large equatorward bias (3.4°) and underestimation of its strength (-1.4 m.s$^{-1}$). The agreement for WMPOS and WMSTR is much better in ARP-AMIP-AC, and remaining errors are insignificant (p < 0.05). The annual variabilities of WMPOS and WMSTR decrease in ARP-AMIP-AC with respect to AMIP. This is beneficial, because ARP-AMIP shows a larger variability in WMPOS than what is found in the ERA-I reanalysis.





**Table 2.** Relative seasonal root mean square error reduction $\Delta_r E$ (in %) south of 20°S with respect to ERA-Interim for ARP-AMIP-AC with respect to ARP-AMIP during the 1981-2010 period for different surface and tropospheric variables at constant pressure levels :

| *Simulations* | JJA | SON | DJF | MAM |
|---|---|---|---|---|
| **SLP** | 87 | 69 | 52 | 75 |
| **T200** | 95 | 93 | 90 | 86 |
| **T500** | 95 | 95 | 96 | 95 |
| **T700** | 88 | 90 | 88 | 87 |
| **T850** | 95 | 93 | 90 | 86 |
| **Z500** | 93 | 87 | 77 | 90 |
| **Z850** | 90 | 71 | 58 | 87 |
| **Q500** | 13 | 14 | 81 | 23 |
| **Q850** | 42 | 7 | -57 | -7 |

### 3.1.2 Atmospheric General Circulation : daily variability

The relative frequencies for each BMU for AMIP, ARP-AMIP-AC and ERA-I are presented in Fig. 3. ARP-AMIP-AC simulation shows a clearly better distribution of daily SLP with reduced RMSE compared to ARP-AMIP and a much better correlation with the ERA-I distribution. The BMU frequencies in ARP-AMIP are generally overestimated for circulation patterns with a

low meridional pressure gradient and low pressures center located relatively far off the Antarctic coasts (1, 2, 3, 6, 11) and/or pressure ridges over the Pacific sector (1, 2, 3, 5). Conversely, frequencies of patterns with large meridional pressure gradient and low pressure systems closer to the Antarctic coasts are mostly underestimated in AMIP. These errors are consistent with the biases evidenced in the analysis of the errors of the mean state in previous paragraph. For ARP-AMIP-AC, although it is also clearly the most frequent pattern in ERA-I, there is a large overestimation of the 20[th] BMU. The large overestimation of

pattern 20 probably reflects the fact that a certain number of circulation patterns present in ARP-AMIP-AC are not correctly represented in the 20 BMU derived using the daily SLP from all simulations. BMU 20 represents synoptic situations with very high meridional pressure gradient and strongly zonal circulation. As indicated by its position at the fringe of the figure, it represents patterns that are in this sense extreme among the situations appearing in the ARP-AMIP simulation, and it apparently best represents the presumably even more zonal circulation patterns with stronger meridional gradients only present in

ARP-AMIP-AC.

### 3.1.3 Near-surface temperatures

In this section, near-surface air temperatures ($T_{2m}$) from ARPEGE simulations are compared with those from a MAR RCM simulation forced at its lateral boundary by ERA-I (Agosta et al., 2019) and those from weather station of the READER data base Fig. 4a. In this analysis, weather stations for which less than 80 % of valid data were available for the season and the

period considered have been discarded from the analysis. The same figure for the ARP-AMIP simulation already presented in

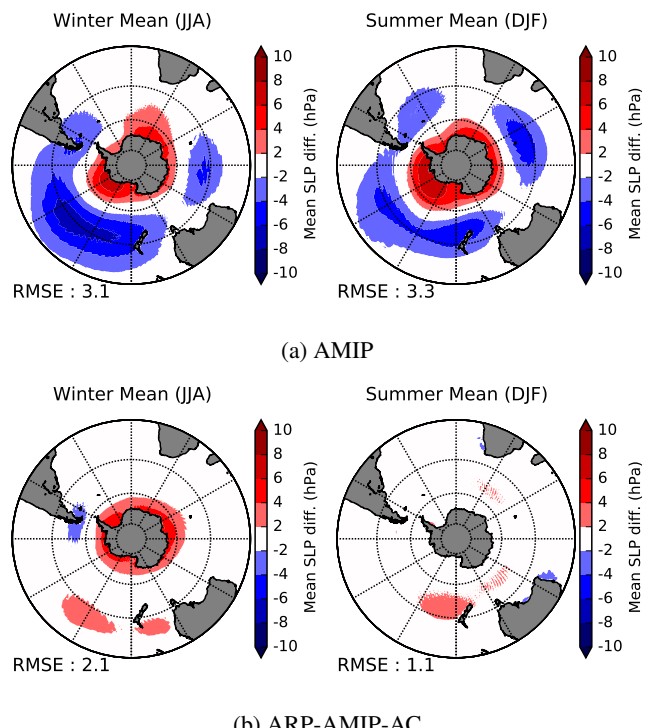

(a) AMIP

(b) ARP-AMIP-AC

**Figure 1.** Difference between ARP-AMIP and ARP-AMIP-AC simulations with ERA-I mean SLP for the reference period 1981-2010 in winter (JJA, *left*) and summer (DJF, *right*). Value of the RMSE are given below the plots.

**Table 3.** Mean annual 850 hPa westerly maximum strength (WMSTR) and position (WMPOS) in ERA-Interim, ARP-AMIP and ARP-AMIP-AC ± one standard deviation of the annual mean. Differences significant at p=0.05 with respect to ERA-I are presented in **bold**.

| Simulation/Data set | WMSTR (m.s$^{-1}$) | WMPOS (°) |
|---|---|---|
| ERA-I | 12.5±0.6 | -51.4±0.8 |
| ARP-AMIP | **11.1±0.5** | **-48.0±1.4** |
| ARP-AMIP-AC | 12.2±0.3 | -51.0±0.6 |



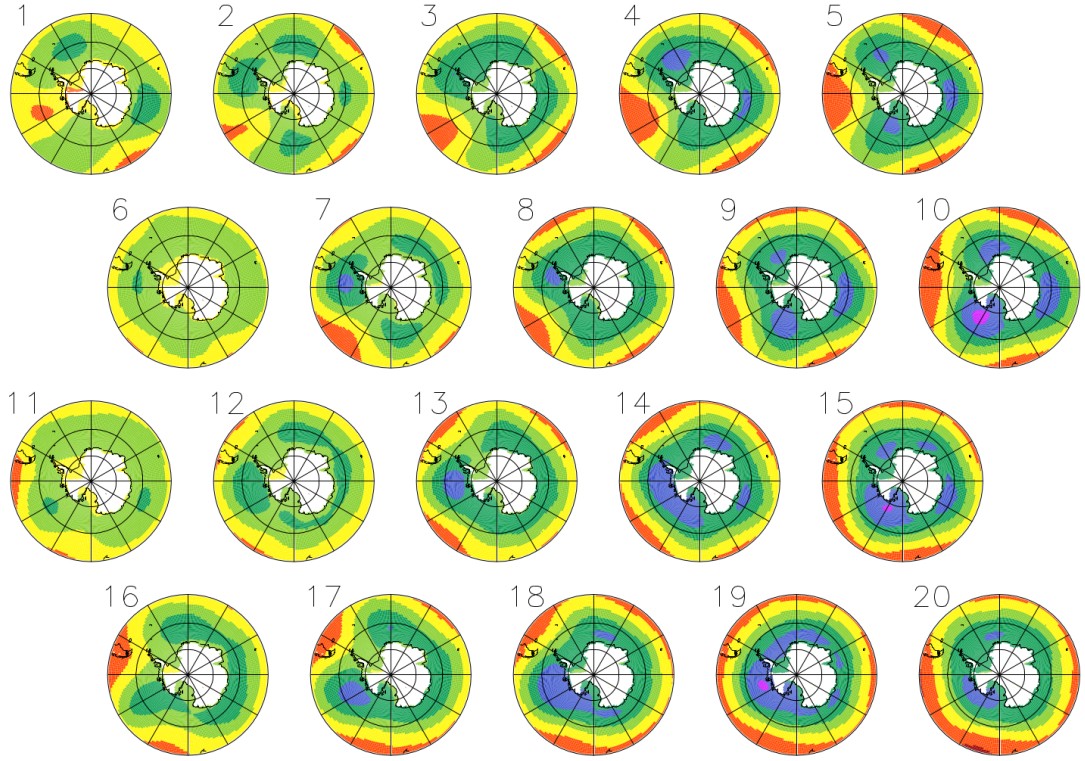

**Figure 2.** Mean Sea Level Pressure (SLP) map for the twenty best matching unit (BMU) obtained after a self-organizing map analysis on daily SLP fields. SLP ranges from 1030 hPa (red) to 960 hpa (purple) with 10 hPa intervals.

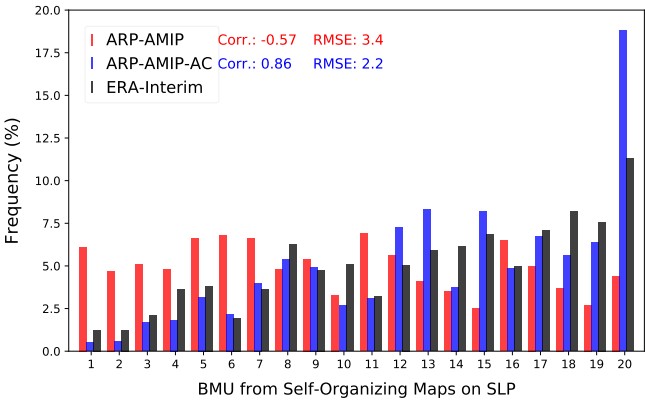

**Figure 3.** Best matching unit (BMU) relative frequency (%) of ARP-AMIP (red), ARP-AMIP-AC (blue) and ERA-I (black) on daily SLP map over the 1981-2010 period. Root mean square errors and pearson correlation coefficient with respect to ERA-I BMU frequencies are shown to the right of the legend.



Beaumet et al. (2019a) can be seen in the supplementary material (Fig. B1). The errors for each station and each season, as well as the mean error and RMSE per regions are also presented in the supplementary material for ARP-AMIP-AC (Table B1) and ARP-AMIP (Table B2). The effect of the atmospheric correction on ARPEGE mean $T_{2m}$ can be seen in the ARP-AMIP-AC minus ARP-AMIP difference (Fig. 4b).

On the East Antarctic Plateau (EAP), the impact of the atmospheric bias-correction is a winter warming (1-3 °C) over large parts of the centre of the Plateau. The warm bias with respect to MAR during this season increases, which is confirmed for instance by a decrease of ARPEGE skills at Vostok in all season but summer (see Table B2).

For coastal East Antarctic READER stations, the cold bias present in every season, particularly in winter, is greatly reduced in ARP-AMIP-AC. The improvement is even dramatic for some stations in eastern East Antarctic (e.g.: McMurdo, Dumont D'Urville, Casey, Davis). The effect of the bias correction is also a cooling of some margins of the eastern East Antarctic Plateau in summer, where the warm bias with respect to MAR decreases. However, the errors remain substantial and significant (p<0.05) in many stations and seasons, especially in the winter (mean error ≈ -2 °C). No improvement is found for the warm bias on the ice shelves and coastal regions of Dronning Maud Land.

Over West Antarctica and the Peninsula, the effect of the atmospheric bias-correction is a warming over much of coastal and central West Antarctica and on the southern and western parts of the Peninsula in winter. In summer, this warming is restricted to the south-western part of the Peninsula, while there is a cooling of the eastern coasts, particularly marked on the Larsen Ice Shelf. The systematic cold bias with respect to MAR and READER stations is greatly reduced, with the largest improvements found in the southernmost stations (Rothera and Faraday). However, the errors remain significant in summer (mean error≈-1.5 °C). Moreover, ARP-AMIP-AC is cold biased with respect to MAR over the Larsen Ice Shelf in summer, which was not the case of ARP-AMIP.

Finally, no substantial improvement is to be reported for the stations located on the islands of the Southern Oceans where the skill of the AMIP-stile control run was already fairly high.

### 3.1.4 Surface mass balance and precipitation

The SMB and its components integrated over the whole grounded ice sheet (GIS) for the 1981-2010 reference period are presented in Table 4 for the two historical ARPEGE simulations presented in this study. The table also displays the correspeding values for ERA-I driven simulations with MAR and RACMO2 (Agosta et al., 2019; van Wessem et al., 2018). The total precipitation over the GIS significantly decreases in ARP-AMIP-AC and is about 10% lower than the estimates from the two polar RCMs, whereas total precipitation in ARP-AMIP is in good agreement with the latter. The integrated GIS SMB is about four times the inter-annual variability uncertainty range ($\sigma$) lower in ARP-AMIP-AC than in MAR-ERA-I and RACMO2-ERA-I as a consequence of lower precipitation and largely overestimated surface sublimation rates (and, to a lesser extent, run-off). In the atmosphere-corrected simulation, there is a significant decrease of run-off and surface snow sublimation with respect to previously uncorrected simulation, yet insufficient to match with the estimates from the two polar-oriented RCMs. The overestimation is still large for the comparison with RACMO2 simulation where sublimation include surface and blowing

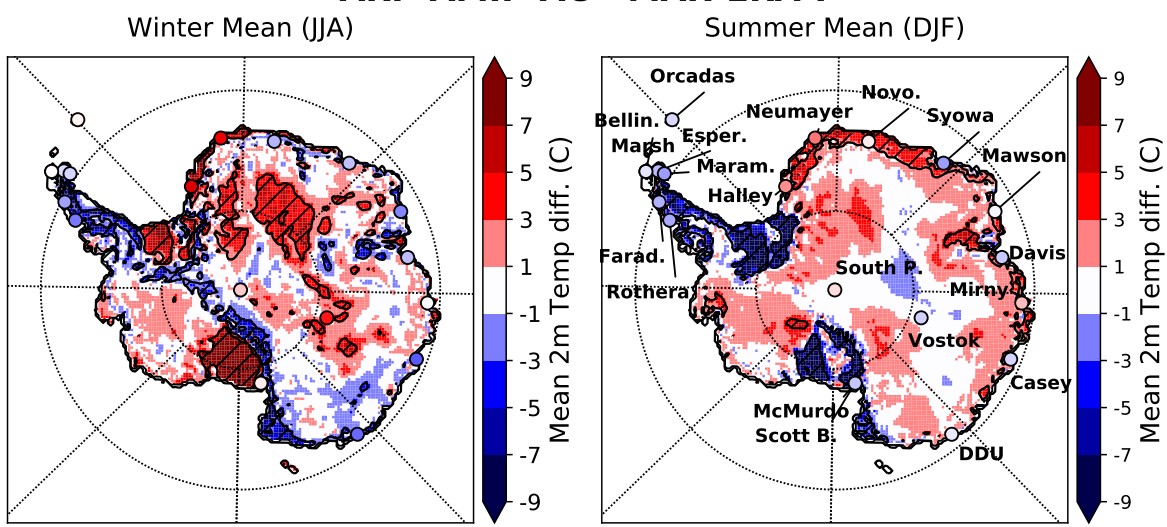

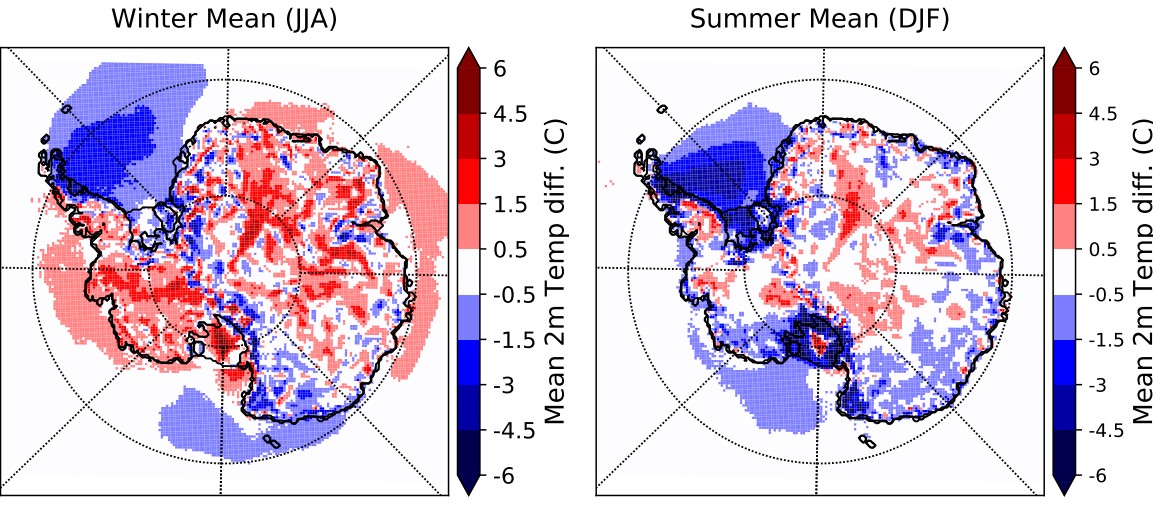

**Figure 4.** (a) ARP-AMIP-AC minus MAR-ERA-I $T_{2m}$ in winter (*left*) and summer (*right*). Circles represent mean bias for weather stations from the monthly READER data base. *Black contour lines* represents where the difference is one standard deviation of MAR $T_{2m}$. (b) Same as (a) but for ARP-AMIP-AC minus ARP-AMIP.





| Simulation | SMB | Precip | Subli. | Run-Off | Rain | Melt |
|---|---|---|---|---|---|---|
| ARP-AMIP | 1970±96 | 2268±94 | 277±17 | 22±14 | 10±2 | 52±32 |
| ARP-AMIP-AC | **1758±119** | **1994±117** | **222±11** | **14±4** | **5±2** | 50±18 |
| MAR-ERA-I[1] | 2158±106 | 2260±104 | 84±10 | 3±2 | 16±3 | 45±15 |
| RACMO2-ERA-I[1,2] | 2117±92 | 2268±99 | 136±4 | 2±2 | 3±1 | – |
| Vaughan et al. (1999) | 1811 | – | – | – | – | – |

**Table 4.** Mean GIS Surface Mass Balance and its components (Gt yr$^{-1}$) ± one standard deviation of the annual value for the reference period 1981-2010. ARPEGE values are integrated over the original model grid and take into account the model land-mask. ARPEGE, [1]MAR and RACMO2 ERA-Interim driven, statistics for 1981-2010 for the Antarctic GIS using MAR grounded ice-masks (GIS area = 12.37 10$^6$ km$^2$). [2]RACMO2 data original source is van Wessem et al. (2018). Statistics in **bold** for ARP-AMIP-AC are statistically different from ARP-AMIP at p=0.05

snow sublimation (see Table 4). Estimations of surface snow melt in ARP-AMIP-AC are however within the 1$\sigma$ uncertainty range when compared to RACMO2 and MAR.

The spatial distributions of total precipitation, surface snow sublimation and SMB of ARP-AMIP-AC are compared with MAR-ERA-I in Fig. 5. The same comparison for the ARP-AMIP simulation was already presented in Beaumet et al. (2019a) and is reproduced in the supplementary material here for convenience (Fig. C1). The effect of the in-line correction of errors on atmospheric general circulation on ARPEGE Antarctic precipitation can be seen in Fig. 6. It shows mainly a drying in many parts of Antarctica such as Marie-Byrd Land, Dronning Maud Land, Victoria Land and the Transantarctic Mountains, a large part of the East Antarctic Plateau up to Adélie Land, and the eastern side of the AP. Conversely, precipitation increases in ARP-AMIP-AC over central and western West Antarctica and over the western side of the Antarctic Peninsula. These changes in precipitations result in a better agreement for the spatial distribution of precipitation with MAR-ERA-I over large parts of West Antarctica, Dronning Maud Land and the Peninsula. However, the disagreement between the two model is still considerable (> 20%) in many places, and the dry bias with respect to MAR-ERA-I present in ARP-AMIP over the East Antarctic Plateau and the Transarctic Mountains increases. Moreover, the agreement in terms of statistics (mean error and RMSE) does not show the increase in skills expected for ARP-AMIP-AC. Therefore, we also compared precipitation in the ARPEGE simulation with those from RACMO2 and assessed the impact of the atmospheric correction on the the comparison between the two models (Fig. 7a and Fig. 7b). Many differences between ARP-AMIP with RACMO2 are the same as those seen in the comparison with MAR, and these are also improved in ARP-AMIP-AC. ARP-AMIP-AC and RACMO2 agree remarkably well (errors below 20%) in many areas with rather complex topography such as Dronning Maud Land, coastal West Antarctica or the Transantarctic Mountains. The systematic dry bias with respect to MAR over the Transantarctic Mountains and Victoria Land is not found in the comparison with RACMO2. The widespread dry bias in ARPEGE over the eastern part of the East Antarctic Plateau and the ridges of the western parts of the Plateau is confirmed in the comparison with RACMO2 and explains most of the ~10% precipitation deficit at the continental scale in the ARP-AMIP-AC simulation with respect to both RCMs.



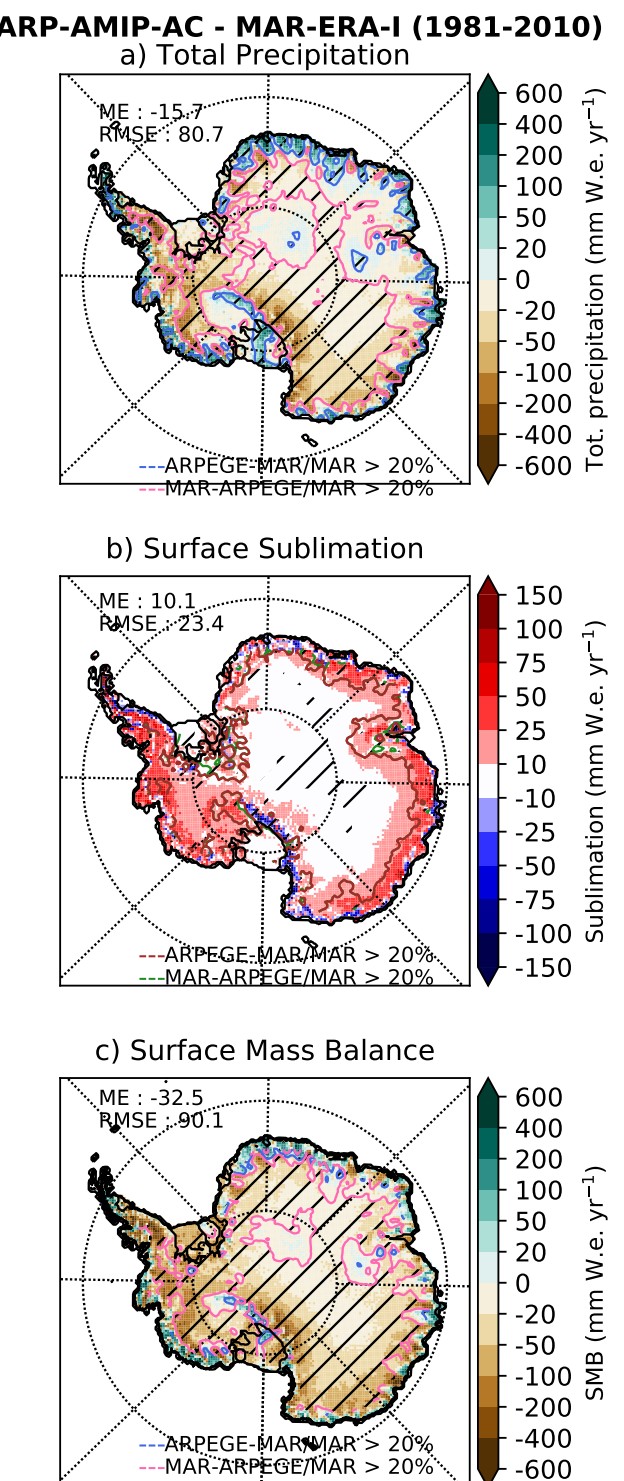

**Figure 5.** Total precipitation (*a*)), surface snow sublimation (*b*)) and SMB (*b*)) for ARP-AMIP-AC minus MAR-ERA-I (mm W.e. yr$^{-1}$) for the reference period 1981-2010. Pink (brown) and blue (green) contour lines represents areas where ARPEGE-MAR differences are larger than 20%. Mean error (ME) and RMSE statistics (mm .W.e yr$^{-1}$) are presented in upper-left corner.


**ARP-AMIP-AC - ARP-AMIP Total Precipitation**

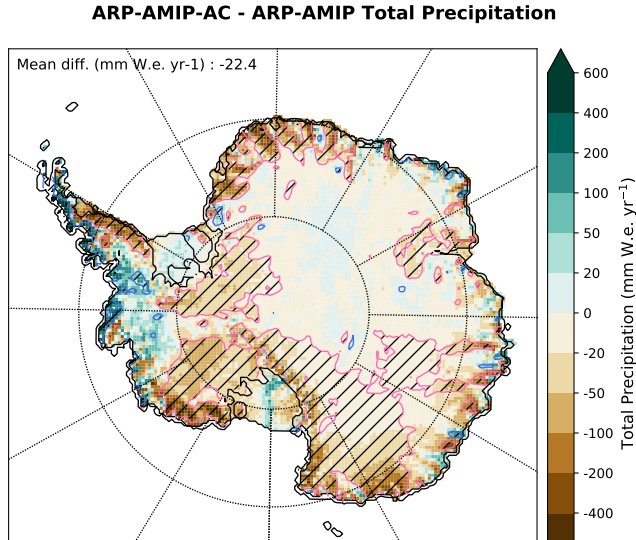

**Figure 6.** Total Precipitation (mm W.e. yr$^{-1}$) difference for ARP-AMIP-AC minus ARP-AMIP for the reference period 1981-2010. Hatched blue (resp. pink) contour lines represents areas where precipitation amounts are 20% larger (lower) in ARP-AMIP-AC.

## 3.2 Climate change signal

In this section, we present and compare the climate change signals for late 21$^{st}$ century obtained in the different ARPEGE RCP8.5 projections presented in this study. Climate change signals are obtained by computing the difference with their reference simulation in present-day climate (see Table 1). Differences in changes in atmospheric general circulation, near-surface temperature, precipitation and SMB obtained when using atmospheric bias-correction are more specifically emphasized.

### 3.2.1 Atmospheric General Circulation

Changes in atmospheric general circulation are summarized by representing the Southern Hemisphere latitudinal profiles of sea-level pressure for each of the present-day simulation and future projections (Fig. 8). The simulated climate change signal is represented by the difference between each projection (coloured lines) and their reference simulation for present-day climate (dashed or plain lines). It can be seen that each future projection is characterized by a strengthening of the mid-latitude highs and a deepening of the circum-antarctic troughs and by a poleward movement of these features with respect to their reference historical simulation. This corresponds to an increasingly positive phase of the Southern Annular Mode (SAM) index, the main mode of variability of atmospheric general circulation in the southern high-latitudes, which is in good agreement with the generally expected consequences of the increase in GHG concentration on the Southern Hemisphere's high latitude atmospheric





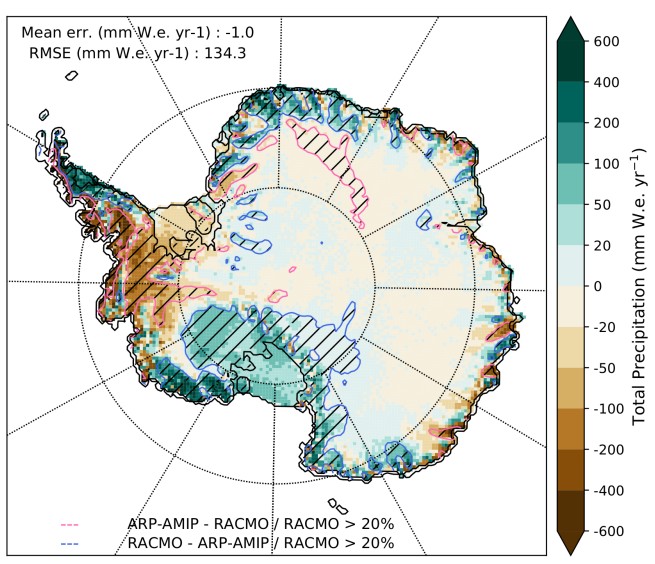

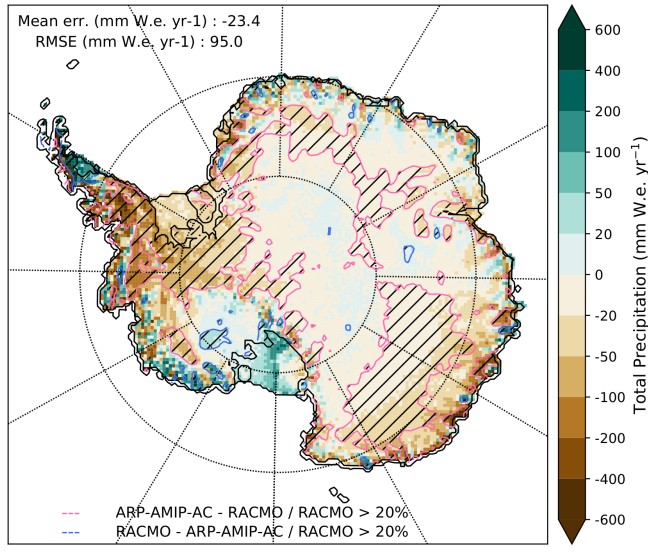

**Figure 7.** (a) ARP-AMIP minus RACMO2-ERA-I total precipitation (mmWe yr$^{-1}$) for the reference period 1981-2010 (b) same as (a) but for ARP-AMIP-AC minus AMIP. Mean error (ME) and RMSE statistics (mmWe yr$^{-1}$) are presented in upper-left corner.

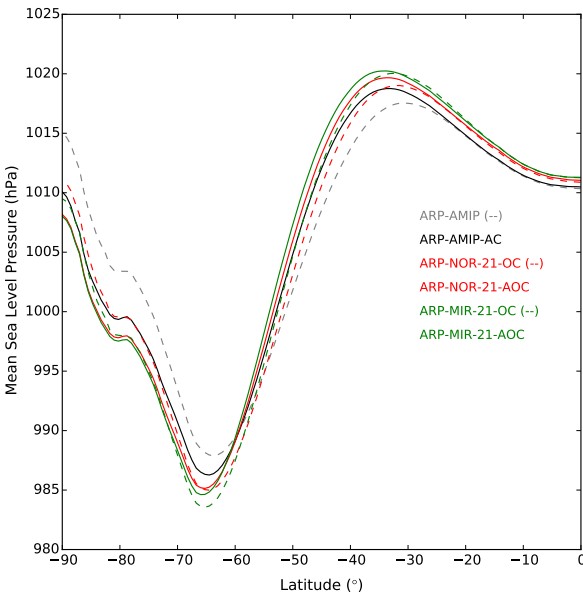

**Figure 8.** Yearly mean Sea-Level Pressure (hPa) for ARPEGE historical simulation (1981-2010) and RCP8.5 projections (2071-2100). Uncorrected control run are displayed in dashed lines and run with atmospheric bias-correction are presented in plain lines. Historical simulations realized with observed SSC are displayed in black (ARP-AMIP-AC) or gray (AMIP). Projections driven by bias-corrected SSC from NorESM1-M (MIROC-ESM) are shown in red (green).

circulation for the late 21[st] century (Arblaster and Meehl, 2006; Miller et al., 2006; Fyfe and Saenko, 2006). However, it is noteworthy that both projections realized with atmospheric bias corrections show much weaker changes in this high pressure increase (resp. low pressure decrease) pattern as well as a weaker poleward shift. This is confirmed by statistics from the changes in 850 hPa westerly maximum strength and position ($\Delta$WMPOS and $\Delta$WMSTR, Table 5). Each future projection
5    displays an increase in WMSTR and poleward movement of the westerly maximum, but the magnitude of these changes are about 50 % weaker in projections realized with atmospheric bias correction. Future projections realized with MIROC-ESM SSC (higher decrease in sea-ice) show a larger southward displacement of the westerlies.

### 3.2.2    Near-surface temperature

10    The increase in mean yearly $T_{2m}$ averaged over the GIS are respectively 3.5$\pm$1.0 and 5.0$\pm$1.3 °C for ARP-NOR-21-AOC and ARP-MIR-21-AOC . This represents respectively a +0.7 and +0.9 °C additional warming (significant at $p < 0.05$) compared to the corresponding projections without atmospheric bias correction. Differences in warming rate per season are presented in Table 6. Differences in warming for the atmospheric-corrected simulations are the largest in summer and are significant for





**Table 5.** Anomalies in Southern westerlies maximum strength ($\Delta$WMSTR, m/s) and position ($\Delta$WMPOS, °) for the different ARPEGE projections

| Simulations | $\Delta$WMSTR (m/s) | $\Delta$WMPOS (°) |
|---|---|---|
| ARP-NOR-21-OC | 1.5 | -2.2 |
| ARP-NOR-21-AOC | 0.8 | -0.8 |
| ARP-MIR-21-OC | 2.0 | -3.8 |
| ARP-MIR-21-AOC | 0.9 | -1.5 |

**Table 6.** Mean seasonal $T_{2m}$ increase (K) for the Antarctic GIS for the different ARPEGE RCP8.5 projection for the late $21^{st}$ century with respect to their historical reference simulation. Anomalies in projections with bias-corrected atmosphere significantly different ($p < 0.05$) from the anomaly in the uncorrected control run level are shown in *bold*.

| Simulations | DJF | MAM | JJA | SON |
|---|---|---|---|---|
| ARP-NOR-21-OC | 3.0±1.4 | 2.6±1.4 | 3.1±1.4 | 2.6±1.0 |
| ARP-NOR-21-AOC | **3.8±1.2** | **3.7±1.2** | 3.3±1.9 | 3.4±1.9 |
| ARP-MIR-21-OC | 3.6±1.5 | 4.6±1.7 | 4.6±1.4 | 3.8±1.5 |
| ARP-MIR-21-AOC | **5.1±1.4** | 5.2±1.7 | 5.1±1.7 | **4.8±2.0** |

both projections, while they are smaller and not significant in winter. For ARP-NOR-21-AOC , the larger warming in autumn (MAM) is also significant, while it is in spring for ARP-MIR-21-AOC .

The spatial distribution of the increase in $T_{2m}$ and corresponding differences in winter (JJA) and summer (DJF) are presented in Fig. 9. The two sets of projections show very similar patterns in terms of differences in regional warming. The larger surface warming in summer in the atmospheric-corrected experiment is essentially the consequence of a stronger temperature increase over East Antarctica. Systematically, near-surface temperature increases are strongest where sea ice is lost. This is particularly noticeable over the northern part of the Weddell Sea. This area (together with Larsen and Ronne-Filchner ice shelves) shows a large additional warming in atmosphere-corrected experiments. In winter, there is a well-marked dipole with lower warming over West Antarctica and the Ross Ice Shelf and higher warming over southern Victoria and Adélie Land.

### 3.2.3 Surface mass balance and precipitation

Absolute and relative increase in SMB and for its different components for the Antarctic GIS are presented in Table7. All projections show an increase in surface mass balance, resulting from the absolute increase in precipitation still being much larger than the corresponding increases in surface sublimation and run-off. This is in agreement with previous Antarctic climate change studies for the late 21st century (e.g., Lenaerts et al., 2016; Krinner et al., 2014; Frieler et al., 2015). Additional increases in total precipitation of +78 and +90 Gt yr$^{-1}$, which corresponds respectively to a +6 and +9 % additional relative increase (not significant at $p < 0.05$), are found respectively for ARP-NOR-21-AOC and ARP-MIR-21-AOC atmospheric corrected projections. In ARP-MIR-21-AOC, significant higher increase in surface sublimation mitigates slightly the increase in SMB



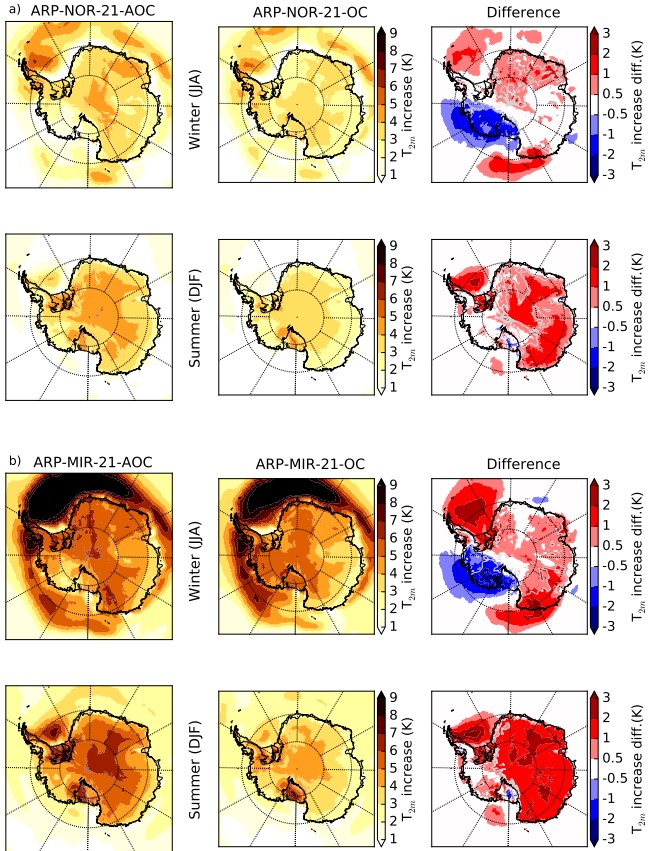

**Figure 9.** $T_{2m}$ anomaly for ARPEGE RCP8.5 projection for the late $21^{st}$ century (reference period : 1981-2010) with atmospheric bias-correction (AOC,*left*), uncorrected atmosphere (OC,*center*) and difference (*right*). Anomalies for winter (JJA) are displayed at the *top* of the subfigures and for summer (DJF) at the *bottom*. Results for projections with bias-corrected SSC from NorESM1-M (resp. MIROC-ESM) are shown in a) (resp. b). *Grey contour lines* is where differences in anomaly is $> 25\%$ with respect to the uncorrected control run.

with respect to ARP-MIR-21-OC . Despite a larger warming at the continental scale, increases and cumulated amounts of precipitation are significantly lower in projections with atmospheric bias correction.

The spatial distribution of precipitation changes for each projection are shown in Fig. 10 along with the differences between the corrected simulations and the uncorrected control runs. Projections with atmospheric bias correction show a smaller precip-
5 itation increase over most of West Antarctica, the Transantarctic Mountains and western Dronning Maud Land. Conversely, the increase in precipitation is larger on the coastal AP, on the Ross side of Marie-Byrd Land and on Adélie and southern Victoria Land, where uncorrected projections suggest a precipitation decrease.





**Table 7.** Absolute values, absolute (Abs.) and relative (Rel.) anomalies for mean SMB and its components (Gt yr$^{-1}$) for the Antarctic GIS in the different ARPEGE RCP8.5 projections (reference period: 1981-2019). Anomalies and absolute values significantly different (p < 0.05) in projections realized with bias-corrected atmosphere with respect to control run are displayed in bold.

| Simulations | SMB | Tot. PCP | Surf. Sublim. | Rainfall | Melt |
|---|---|---|---|---|---|
| **ARP-NOR-21-OC** | **2334±181** | **2742±176** | 331±21 | 27±7 | 184±82 |
| *Abs. anomaly (Gt yr$^{-1}$)* | 364±195 | 474±179 | 55±26 | 17±8 | 132±137 |
| *Rel. anomaly* | 19% | 21% | 20% | 171% | 252% |
| **ARP-NOR-21-AOC** | **2172±143** | **2534±158** | **284±19** | **21±2** | 210±79 |
| *Abs. change (Gt yr$^{-1}$)* | 422±169 | 540±176 | 62±18 | **16±8** | **160±64** |
| *Rel. change* | 24% | 27% | 28% | 320% | 320% |
| **ARP-ARP-MIR-21-OC** | **2637±156** | **3108±202** | 345±29 | 52±15 | 306±144 |
| *Abs. change (Gt yr$^{-1}$)* | **667±202** | **840±227** | 68±23 | 42±15 | 254±118 |
| *Rel. change* | 34% | 37% | 25% | 416% | 484% |
| **ARP-MIR-21-AOC** | **2460±197** | **2903±222** | **308±23** | **45±14** | 359±118 |
| *Abs. change (Gt yr$^{-1}$)* | 709±218 | 909±229 | **86±23** | **40±15** | 309±104 |
| *Rel. change* | 40% | 46% | 39% | 800% | 618% |

## 4 Discussion

### 4.1 Representation of the historical climate

In this section, we discuss following our comparisons with ERA-I reanalyses, *in-situ* observations and polar-oriented RCMs, the improvement in skills associated with atmospheric empirical in-line bias correction of ARPEGE model for the atmospheric
general circulation in the high southern latitudes as well as for the surface climate of the Antarctic ice-sheet. Where relevant, we also discuss possible biases compensation suppression associated the systematic errors correction for atmospheric general circulation. The limitations and potential improvement of the method are also addressed.

### 4.1.1 Large-scale atmospheric circulation

Compared to ERA-I, the simulated large-scale atmospheric circulation South of 20°S using atmospheric bias correction in
ARP-AMIP-AC is dramatically improved with respect to the uncorrected ARP-AMIP control run. Reduction of the seasonal RMSE between 50 to 70 % with respect to ARP-AMIP are found for many mid and upper tropospheric variables. Improvement for the position and strength of the mid-latitude surface westerly jet are also clear and the difference with ERA-I are insignificant in ARP-AMIP-AC. The analysis of the best matching unit frequencies in the self-organizing maps analysis also evidenced much better representation of daily variability in ARP-AMIP-AC. In this regard, it is noteworthy that Guldberg et al. (2005)
also found large (20-30 %) improvement in the seasonal forecasting skills of their corrected model in the Southern Hemisphere while no improvement was reported on average in the Northern Hemisphere.

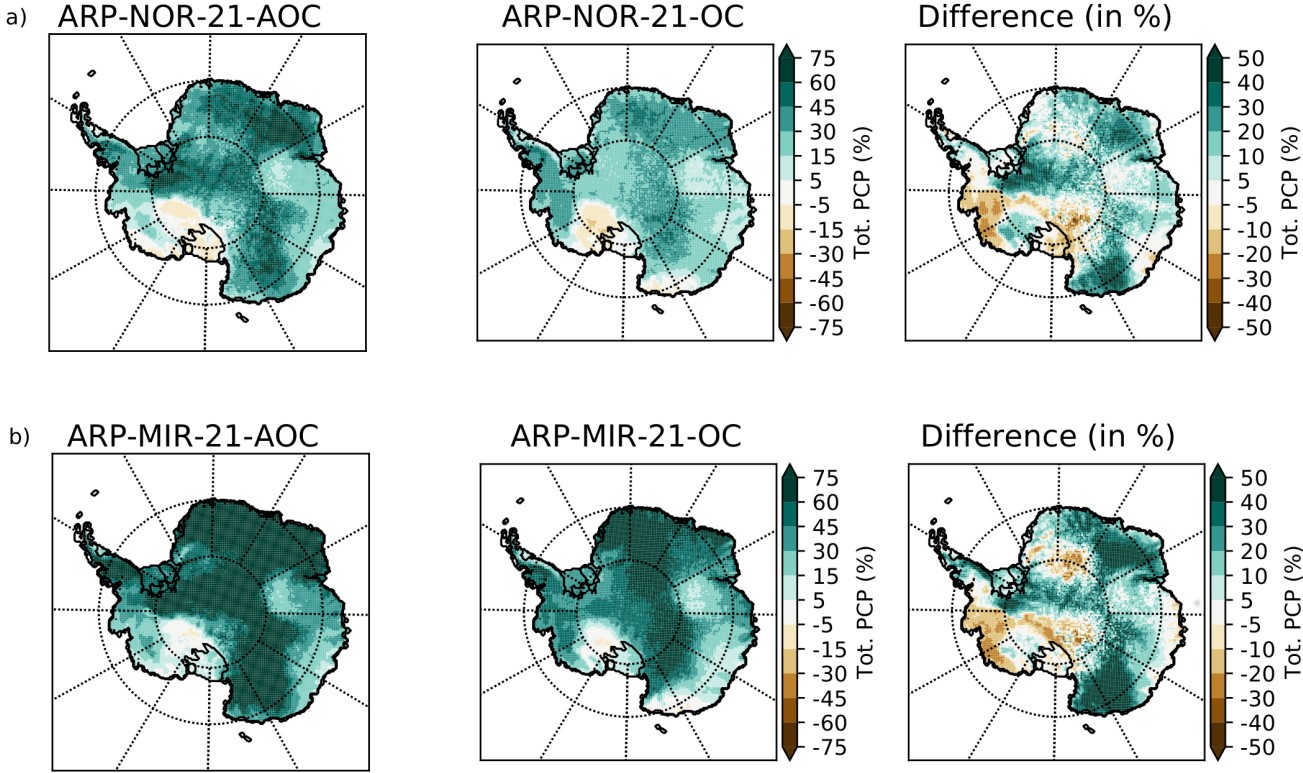

**Figure 10.** Late 21$^{st}$ century anomalies in yearly total precipitation (%) for ARPEGE RCP8.5 projections with atmospheric bias-correction (*left*), uncorrected reference simulation (*center*) and difference (*right*). Results for projections with bias-corrected SSC from NorESM1-M (resp. MIROC-ESM) are presented in a) (resp. b).

In our experiment, we found no bias reduction in tropospheric humidity (Q850 and Q500) and lower tropospheric temperature (T850). As a reminder, 850 hPa is the level at which the correction on the tendency for the atmospheric variables starts to decrease towards the surface. The new bias patterns appearing at this level in ARP-AMIP-AC over southern mid-latitudes land masses and the adjacent oceans (see Section. A) are most likely linked to surface processes (e.g., development of marine

5   stratocumulus, convective boundary layer). The typical time scale for the development of such processes is shorter than the 72 h relaxation time used in the nudged simulation realized to build the correction terms. As a consequence, this advocates for further sensitivity tests using different relaxation times depending on the variable, especially for specific humidity, in order to improve the skills of the corrected model for this variable in future similar experiment. Re-tuning the atmospheric model during the nudging step could also be an option to ensure consistency of the free physical parameters with the modified circulation

10  characteristics.





### 4.1.2 Near-surface temperature

The correction of large-scale circulation errors in ARP-AMIP-AC has clearly lead to a reduced cold bias found in ARP-AMIP control run over the western and southern part of the Antarctic Peninsula, especially in winter. The skill of the model improves dramatically for stations such has Faraday and Rothera and for most stations of the AP in general. Warm and moist advection from the North-West over this part of the Peninsula in indeed underestimated in ARP-AMIP. However, the east coast of the Peninsuala has cooled dramatically in ARP-AMIP-AC and there is now a substantial cold bias with respect to MAR.

The skill of ARP-AMIP-AC also exceeds that of ARP-AMIP for coastal stations of East Antarctica (e.g., Mc Murdo, Scott Base, Casey and Davis). However, for many stations, the cold bias remains substantial and statistically significant. In this perspective, the comparison with McMurdo and Scott Base is interesting. These stations, distant by only 3 kilometers and located at the same altitude, belong to the same ARPEGE grid point. While ARP-AMIP-AC bias with respect to Scott Base is small and statistically insignificant in all season but summer, the cold bias is ~3 °C (significant) throughout the year with respect to Mc Murdo. This example shows the limited spatial representatvity of weather stations in coastal Antarctica, and therefore the comparisons with a 35 km horizontal resolution atmospheric model should therefore be interpreted carefully.

On the East Antarctic Plateau, the warm bias in winter increases in ARP-AMIP-AC in both comparisons with weather station and MAR-ERA-I. This bias increase seems to result from the removal of a bias compensation that was present in ARP-AMIP. We investigated the value of the near-surface temperature inversion in both present-day ARPEGE simulation (not shown) as the difference between surface temperature ($T_S$) and the temperature of the first atmospheric layer, located around 6 to 10 meters height in this ARPEGE setting (level at 0.9988×surface pressure). The value of this near-surface temperature inversion around (-15°C) does not decrease in ARP-AMIP-AC. It even increases locally, suggesting small changes in surface boundary layer processes. However, there was a substantial negative bias with respect to MAR in incoming long wave radiation (LWD) in ARP-AMIP, which disappears in ARP-AMIP-AC due to considerable warming at 500 hPa in winter (Fig. A3). The warm bias in winter with respect to MAR, which generally varies between 3 and 5 °C (similar biases are found when considering READER weather stations or RACMO2 $T_{2m}$) is to be put into perspective with even larger biases sometimes found over the East Antarctic Plateau in climate models or in even climate reanalyses (Dutra et al., 2015; Fréville et al., 2014; Bracegirdle and Marshall, 2012).

The warm temperature bias over Antarctic ice shelves does not decrease in ARP-AMIP-AC, it even increases over Neumayer and Halley stations, especially in winter. ARPEGE model deficiencies for surface temperatures over the large Ronne-Filchner and Ross ice-shelves have been widely adressed in Beaumet et al. (2019a). They are mostly due to misrepresented stable boundary layers processes and excess (with respect to MAR) in long-wave downward radiation in winter associated with higher cloud cover. Large discrepancies have also been evidenced between polar-oriented RCMs MAR and RACMO2 for these areas.





### 4.1.3 Surface mass balance and precipitation

Over the GIS, the atmospheric bias correction caused a decrease of total precipitation of 285 Gt yr$^{-1}$ (∼3 times the inter-annual variability). Unsurprisingly, the precipitation minus evaporation difference (P-E), which is a good approximation of moisture transport, has also decreased, by 221 Gt yr$^{-1}$. The main feature of the atmospheric bias-correction in ARP-AMIP-AC

with respect to ARP-AMIP is an increase of SLP at mid-latitude, and a decrease around Antarctica, which can be seen as a larger dominance of SAM$^{+}$-type patterns (see BMU 15,17,18,19,20 in Fig. 2 and  3). In this regard, we note that changes in precipitation in ARP-AMIP-AC with respect to ARP-AMIP (Fig. 6) bear many similarities with the signature of a positive SAM pattern on Antarctic precipitation assessed in a RACMO2 ERA-Interim driven simulation in Marshall et al. (2017). It is noteworthy that the link between circulation changes induced by the bias correction and ensuing precipitation changes seen in

our simulations is very similar to the effect of bias corrections in the LMDZ model reported by Krinner et al. (2019a).

The correction of atmospheric circulation errors has caused a drying over Dronning Maud and Marie-Byrd Land and an increase in precipitation over central and eastern West Antarctica as well as on the western side of the Peninsula. In these regions, the agreement with MAR and RACMO2 has increased substantially. The improvement of the agreement (RMSE) is much larger for the comparison with RACMO2, and except for some parts the AP and of Adélie Land, the agreement is quite high (errors

below 20 %) for many coastal areas. The comparison with both RCMs also reveals that ARP-AMIP-AC is likely too dry over much of the East Antarctic Plateau. Further investigations are needed in order to identify possible causes of this error, such as deficiencies in moisture transport over the high continental interior or misrepresented clear-sky precipitation processes, which bring a large share of total precipitation in this area (Walden et al., 2003). Improvement for the representation of precipitation, cloudiness and therefore radiative budget for the high continent interior of Antarctica could be obtained for ARPEGE by

accounting for saturation with respect to the ice for the formation of cloud condensates and by tuning cloud microphysics to the cold context of Antarctica such as done in Van Wessem et al. (2014); van Wessem et al. (2018).

The better agreement with precipitation modelled by the polar-oriented RCMs MAR and RACMO2 precipitation increase the confidence in the reliability of spatial distribution of Antarctic SMB modeled in ARP-AMIP-AC. Both RCMs have been widely validated against *in-situ* measurements of Antarctic SMB (e.g., Agosta et al., 2019; van Wessem et al., 2018). The closer

agreement between the ARPEGE ARP-AMIP-AC simulation and RACMO2 in coastal areas offers an interesting opportunity to further investigate causes of remaining disagreement between MAR and RACMO2 in Antarctic precipitation and SMB identified in Agosta et al. (2019). In this paper, it is argued that RACMO2 struggles to represent sublimation of snowfall in the dry katabatic layer present at the ice sheet margins and in valleys such as shown in Grazioli et al. (2017). The same issue could be present in ARPEGE simulations due to the relatively large model physics time step used (15 min). Over the lee-side

of the Transarctic mountains (Victoria Land), the good agreement between ARPEGE and RACMO2 seems to confirm a wet bias in MAR over this area, which has also been identified after comparison with and sparse *in-situ* observations in Agosta et al. (2019).



## 4.2 Climate change signals

In this section, we discuss the differences in projected climate change for atmospheric general circulation of the southern high-latitudes and for Antarctic surface climate between corrected and uncorrected projections and their consistency with previous studies.

### 4.2.1 Large-scale atmospheric circulation

The increase of the surface pressure gradient between mid and high Southern latitudes for the late 21$^{st}$ century is substantially weaker in the atmosphere-corrected projections. The strengthening and the poleward shift of the 850 hPa westerly wind maximum are reduced by about 50 %. This weaker climate change signal for polarward shift in the corrected simulations can be related to the dependence on historical state of projected changes in atmospheric circulation evidenced in Bracegirdle et al. (2013). However, concerning the strength of the westerlies, Bracegirdle et al. (2013) did not find a significant link between the simulated historical state and projected changes. Larger differences in sea-level pressure projected changes found over the Pacific sector in our atmosphere corrected experiment are also consistent with this results from Bracegirdle et al. (2013) who found that the historical state dependence was stronger in this area. In similar experiments conducted with LMDZ model (Krinner et al., 2019a) with different oceanic forcings, a smaller decrease (increase) of the high latitudes low (mid-latitudes highs) pressure is also found. The magnitude of these difference is however much more reduced when compared to the results with ARPEGE. In future works, the less pronounced strengthening of the pressure gradient and the large magnitude of this reduced poleward shift found for atmospheric-corrected ARPEGE experiment however need to be confirmed with supplementary experiments using either other set oceanic surface forcing or other atmospheric models, considering the large impacts of these on the projected climate change for the Southern Hemisphere high-latitudes.

### 4.2.2 Near-surface temperatures

The additional 0.7 to 0.9 °C warming of the Antarctic GIS found in both atmosphere-corrected projections is largely due to the much larger warming of East Antarctica, especially in summer. This is consistent with a weaker increase of the pressure gradient in these projections, as this corresponds to a lower increase towards a more positive phase of the SAM in future climate. The link between negative (positive) anomalies of the SAM and positive (negative) temperatures anomalies over the East Antarctic Plateau has been established in many previous studies (e.g., Marshall and Thompson, 2016; Kwok and Comiso, 2002). However, following this hypothesis and findings from Marshall and Thompson (2016), less pronounced positive phase of the SAM should also result in a larger warming over West Antarctica and a weaker warming over the northern Peninsula in atmosphere-corrected projections, which is not the case here. Nevertheless, we note that in Marshall and Thompson (2016), temperature variability over West Antarctica and over the Peninsula is also strongly linked to the polarities of the first and second Pacific South American modes (PSA1 and PSA2), the second and third leading empirical functions of geopotential heights south of 20°S. Moreover, the much weaker deepening of the Amundsen Sea Low found in our atmosphere-corrected experiments is consistent with weaker increase in winter temperatures over West Antarctica and higher increase over southern





Adélie and Victoria Land. The mean position of the Amundsen Sea Low was indeed shown to be located over the east-side of the Ross Sea in winter (Raphael et al., 2016), and so it is in ARPEGE simulations. The large additional warming found for the surface of the East Antarctic Plateau in summer (+1 to +2 °C) is also found higher up in altitude (500 hPa), which results in increased LWD.

Overall, most of the differences in near-surface temperatures warming found are consistent with corresponding differences in large-scale atmospheric circulation changes and the relation found between pressure and temperature anomalies in previous studies.

### 4.2.3   Precipitation and surface mass balance

Both atmospheric bias-corrected projections suggest a higher, yet not significant, precipitation increase of respectively +6 and
+9 % in ARP-NOR-21-AOC and ARP-MIR-21-AOC. There is an additional increase in moisture transport towards the AIS (approximated through P-E) of +3.5 and +5.8 % respectively with respect to uncorrected control run. The sensitivity to temperature of the increase in precipitation ($\alpha$) in ARP-NOR-21-AOC and ARP-MIR-21-AOC are respectively +7.7%.K$^{-1}$ and +9.1%.K$^{-1}$, whereas these were respectively +5.2% and +8.8% in the uncorrected control run. This suggests that a large part of the additional increase in precipitation is due to different changes in the atmospheric general circulation, particularly in the
projection driven by NorESM1-M oceanic boundary conditions, while a remaining part is of course due to increased moisture holding capacity of the atmosphere. The latest values found for $\alpha$ are somewhat higher than previous estimates (Krinner et al., 2008; Frieler et al., 2015; Ligtenberg et al., 2013; Palerme et al., 2017), which usually range between +5 and +7 %.K$^{-1}$. However, Bracegirdle et al. (2015) also evidenced that $\alpha$ tends to be higher in models projecting a larger decrease in sea-ice extent, which is the case for experiments forced by MIROC-ESM sea-ice anomaly.

In addition to this, Palerme et al. (2017) found that CMIP5 models which agree better with Antarctic snowfall derived from CloudSat show a larger warming (+0.4 °C) and a higher precipitation increase (+4.8 %) with respect to the ensemble mean in their RCP8.5 projections. Therefore, it would be interesting to investigate whether models agreeing with CloudSat snowfall are doing so because of a better representation of the atmospheric general circulation.

At the regional scale, the higher precipitation increase over Adélie and southern Victoria Land, and the corresponding weaker increase over most of West Antarctica can be related, as for the corresponding dipole in differences of warming in winter, to the weaker deepening to the Amundsen Sea Low. Winter is indeed, with autumn, the season of highest precipitation rates over peripheral Antarctica (Palerme et al., 2017).

### 4.3   Implication of bias-correction and perspectives

The large bias reductions for large-scale atmospheric circulation and surface climate obtained in this study should not be perceived as an argument against pursuing the efforts to improve the dynamics and physics of coupled and atmospheric models in a physically consistent and comprehensive way, nor as a loss of confidence in these tools. These are crucial in order to explore some feedbacks and interactions between the different component of the Earth system in a warming climate. Yet, as long as



biases of state-of-the-art climate models are of about the same order of magnitude as projected changes at the end of current century (Flato et al., 2013), *a posteriori* statistical bias-correction (Hall, 2014; Maraun and Widmann, 2018) will be applied to future projections before they are used as input for impact studies assessment. However, these methods fail to correct for biases due to atmospheric circulation errors (Eden et al., 2012; Stocker et al., 2015; Maraun et al., 2017). Therefore, the method

presented in this study offers an excellent opportunity to circumvent this drawback. Bias stationarity is a strong hypothesis needed to support the application of run-time atmospheric bias-correction in climate projections. However, the evidence of large stationarity in biases of coupled models on large-scale atmospheric circulation evidenced in Krinner and Flanner (2018) supports this application under strong climate change.

Applying this bias correction method also allows to assess remaining uncertainties on projected climate change in coupled

and atmospheric climate models, and it could help to identify which efforts one should focus on in order to reduce these uncertainties. Because the significant differences in large-scale atmospheric circulation and surface climate changes reported in this study could have large impacts on projected changes of the Antarctic ice-sheet mass balance, we suggest to use surface forcing provided by our new projections to drive ice-dynamics or ocean and ice shelves interactions studies. Downscaling the projections presented in this study with polar oriented RCMs such as MAR and RACMO2 could also help to identify

remaining uncertainties associated with ARPEGE biases on Antarctic surface climate and SMB mostly associated with its less sophisticated surface snow scheme.

## 5    Summary and Conclusion

In this study, we used empirical run-time bias correction following the method described in Guldberg et al. (2005) or in Kharin and Scinocca (2012). In order to build correction terms, we used the climatology of the adjustment term on tendency errors

coming from an ERA-Interim driven ARPEGE simulation over the 1993-2010 period. In this experiment, nudged variables were air temperatures, air specific humidity, logarithm of surface pressure, divergence and vorticity with a relaxation time of 72 hours.

The application of this method over present climate (1981-2010) yielded a substantially improved large-scale atmospheric circulation in the Southern Hemisphere. The biases of westerly wind maximum position and strength were almost completely

suppressed. This improvement of Southern Hemispheric general circulation produced a decrease of the biases on near-surface temperature over the Antarctic Peninsula, while we found a slightly increased warm winter bias on the East Antarctic Plateau. Regarding precipitation, the agreement with polar-oriented RCMs MAR and RACMO2 has increased, especially in the comparison with the latest, where differences below 20 % are reported over most coastal areas. A dry bias in the atmosphere-corrected experiment over the summit of the East Antarctic Plateau was also evidenced by this comparison with polar-oriented RCMs.

The application of the method for future climate projections (RCP8.5) using bias-corrected oceanic forcing from MIROC-ESM and NorESM1-M has revealed considerable differences in projected changes in large-scale atmospheric circulation. The strengthening and the poleward shift of the westerly wind maximum are reduced by about 50% with respect to uncorrected reference projections. These differences in change of atmospheric general circulation have caused significant additional warm-





ing of +0.7 to 0.9 K, resulting essentially from the much larger warming of East Antarctica in summer. A dipole with higher warming and increase in precipitation over southern Victoria and Adélie Land and corresponding lower warming and increase in precipitation over most of West Antarctica is also found in winter. This is attributed to a reduced deepening of the Amundsen Sea Low in the atmosphere-corrected projections.

The magnitude of the difference in changes of large-scale atmospheric circulation needs to be confirmed with experiments using other oceanic forcing or other atmospheric models, as these would have large impacts on the evolution of the Antarctic ice-sheet mass balance. However, a reduced poleward shift of the westerlies maximum in the bias-corrected experiments is consistent with the state dependence on historical biases in CMIP5 coupled and atmospheric models projections evidenced by Bracegirdle et al. (2013). Many of the differences found in projected changes in temperature and precipitation are also

consistent with previously evidenced signatures of pressure anomalies, especially considering a weaker SAM+ anomaly (Kwok and Comiso, 2002; Marshall and Thompson, 2016; Marshall et al., 2017).

Because statistical bias corrections, usually applied to climate model output before their use for impact studies, generally fails to correct biases associated with errors on atmospheric general circulation (e.g., Hall, 2014; Maraun et al., 2017; Maraun and Widmann, 2018), the method proposed in this study offers interesting perspectives. The downscaling of atmosphere-corrected

projections with polar-oriented RCMs could help to better constrain the evolution of the future Antarctic ice-sheet surface mass balance and evaluate remaining uncertainties in this study associated with ARPEGE biases on surface climate. The potentially large effect on the Antarctic ice-sheet of the differences in snow accumulation and surface climate changes suggested in this study should be explored using surface forcings coming from atmosphere-corrected projections to drive ice-dynamics or ocean and ice shelves interactions impact studies.

*Data availability.*  Historical run and future projections presented in this study are available on Antarctic Cordex grid at daily time step for atmospheric surface temperature (mean,min and max), precipitation, snowfall, surface snow melt, surface run-off and surface snow sublimation at the following address : https://zenodo.org/record/4059193#.X4mUgCU6_RZ (DOI :10.5281/zenodo.4059193). Data and metadata mostly follow Cordex recommendations.

*Competing interests.*  The authors have no competing interests.

*Acknowledgements.*  This publication was supported by PROTECT. This project has received funding from the European Union's Horizon 2020 research and innovation programme under grant agreement No 869304. This is PROTECT contribution number XX.
We acknowledge the World Climate Research Programme's Working Group on Coupled Modelling, which is responsible for CMIP, and we thank the climate modeling groups participating to CMIP5 for producing and making available their model output. For CMIP the U.S. Department of Energy's Program for Climate Model Diagnosis and Intercomparison provides coordinating support and led development of

software infrastructure in partnership with the Global Organization for Earth System Science Portals.



The Centre National de Recherches Météorologique (Météo-France, CNRS) and associated colleagues are warmly thanks for providing resources and help to run ARPEGE model.

We also thank the Scientific Committee on Antarctic Research, SCAR and the British Antarctic Survey for the availability of the SCAR-READER data base.

5  We kindly acknowledge Michiel van den Broeke for providing RACMO2 data and helpful discussions.

*Author contributions.*  AA prepared ARPEGE version and scripts that were used to produce ARPEGE runs used in this study. MD performed the sensitivity tests and ran the bias-corrected simulations. JB ran the non-corrected reference runs, analysed the output and prepared the manuscript and the figures. CA provided MAR outputs and usefull scripts for the interpolation and comparisons of MAR, RACMO2 and ARPEGE output. GK read and corrected the manuscript many times. All the authors have provided complete reviews and relevant remarks

10  for the improvement of the manuscript.



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

## Appendix A: Large-scale atmospheric circulation

In this section, we present some of the results mentioned in the results or discussion section for large-scale atmospheric circulation in order to facilitate the comprehension of the discussion. In Fig. A1, we can see the large bias-reduction with respect to ERA-I for 200 hPa temperatures in ARP-AMIP-AC. However, we can see the slight increase in spring and winter of the warm bias over the South Pole already present in spring for AMIP.

In Fig. A2, we can see the remaining bias on 850 hPa temperatures in ARP-AMIP-AC with respect to ERA-I. The bias is close to zero in most places except for relatively small warm bias (∼ 1-2K) over mid-latitudes land masses (South America, South Africa and Australia). Wet or dry biases are found over the same places in 850 and 500 hPa specific humidity, but their sign vary depending on the season or the level considered (*figures not shown*). These biases were absent in ARP-AMIP (*figures not shown*) and probably results from errors on planetary boundary layer or clouds processes (i.e. convection).



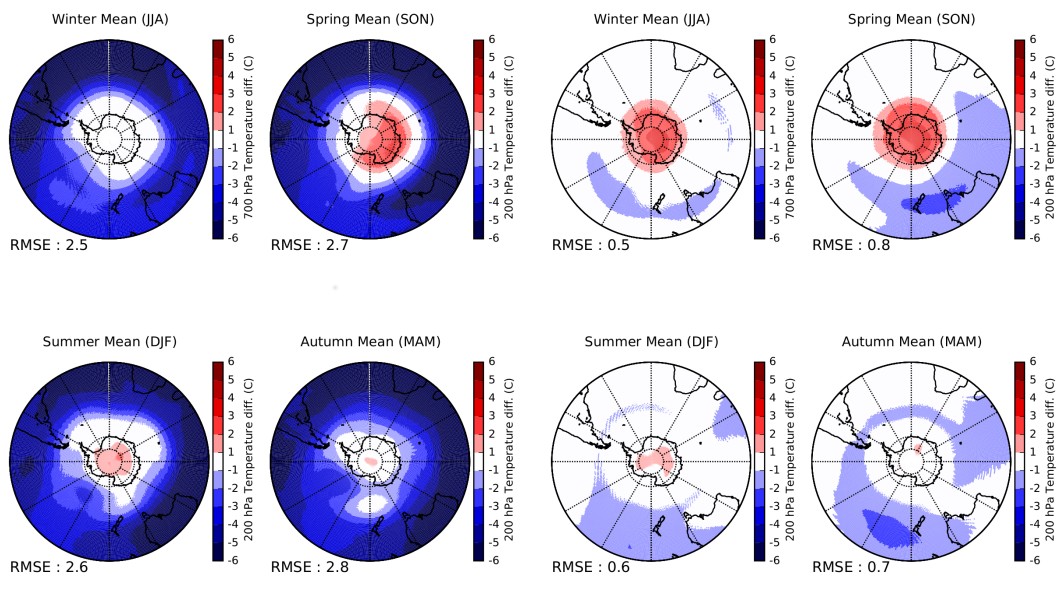

(a) ARP-AMIP - ERA-I                    (b) ARP-AMIP-AC - ERA-I

**Figure A1.** Seasonal ARP-AMIP (top) and ARP-AMIP-AC (bottom) bias on 200 hPa temperature (K) with respect to ERA-Interim over 1981-2010.

**Table A1.** Relative seasonal root mean square error reduction $\Delta_r E$ (in %) south of 20°S with respect to ERA-Interim for ARP-AMIP-AC with respect to ARP-AMIP during the 1981-1992 period for different surface and tropospheric variables at constant pressure levels :

| Simulations | JJA | SON | DJF | MAM |
|---|---|---|---|---|
| **SLP** | 78 | 55 | 48 | 68 |
| **T500** | 91 | 93 | 94 | 89 |
| **Z500** | 87 | 83 | 72 | 81 |
| **Q500** | 3 | 1 | 77 | -1 |

In Fig. A3, we can see that there is substantial warming in winter 500 hPa temperatures in ARP-AMIP-AC with respect to AMIP. This warming resulted in an increase downward longwave radiation over the East Antarctic Plateau and explains the increase of the winter warm bias in this area in near-surface temperatures in ARP-AMIP-AC.





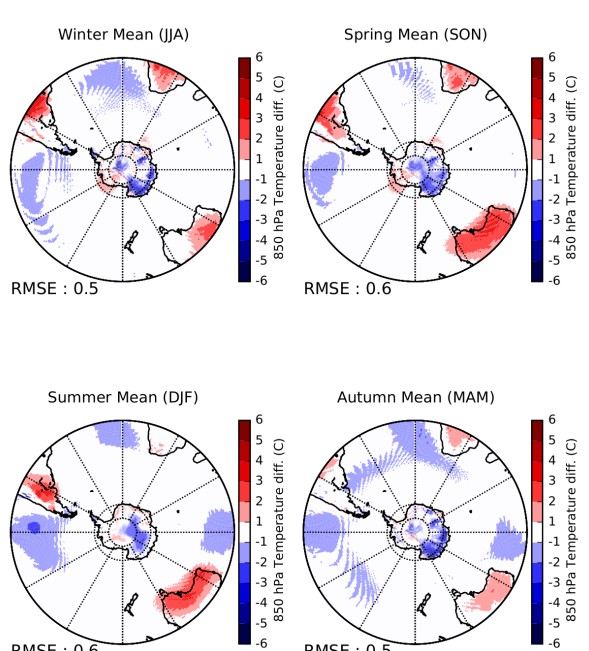

**Figure A2.** Seasonal ARP-AMIP-AC bias on 850 hPa temperatures with respect to ERA-I over 1981-2010.

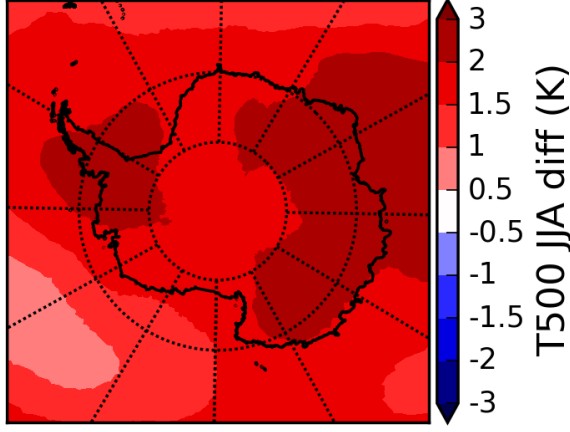

**Figure A3.** ARP-AMIP-AC - ARP-AMIP winter (JJA) 500 hPa temperatures


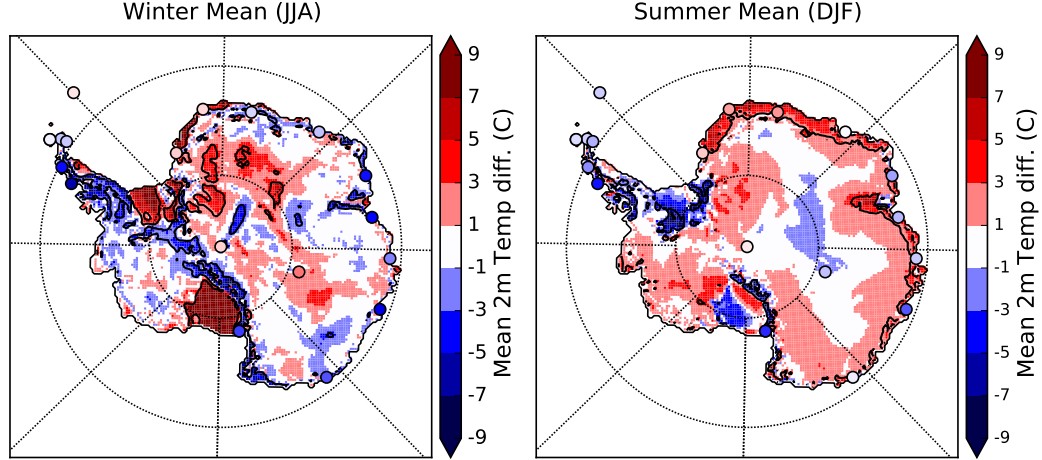

**Figure B1.** ARP-AMIP minus MAR-ERA-I $T_{2m}$ in winter (*left*) and summer (*right*). Circles represent differences with stations from the monthly READER data base. *Black contour line* represents where the difference is one standard deviation of MAR $T_{2m}$.

## Appendix B: Near-Surface Temperatures

In this section, we present the difference with ERA-I driven MAR simulation in $T_{2m}$ for the ARP-AMIP simulation (Fig. B1, already shown in Beaumet et al. (2019a)). The differences with *in-situ* stations from READER data base are presented in Tab B1 for ARP-AMIP-AC and ARP-AMIP Tab B2. Statistics are gathered per Antarctic regions.

## 5  Appendix C:  Surface Mass Balance

In this section, we present the ARP-AMIP comparison with MAR ERA-I for total precipitation, surface sublimation and surface mass balance (Fig. C1) already presented in Beaumet et al. (2019a). The similar comparison for ARP-AMIP-AC is presented in Section 3.1 (Fig. 5).



**Table B1.** Error on READER weather station $T_{2m}$ in the ARP-AMIP-AC simulation for the reference period 1981-2010. Errors significant at $p < 0.05$ are presented in **bold**.

| Stations | DJF | MAM | JJA | SON |
|---|---|---|---|---|
| **EAP** | | | | |
| Amundsen Scott | 0.76 | **3.15** | 1.22 | 0.7 |
| Vostok | **-1.12** | **4.45** | **5.44** | **2.24** |
| *Mean error* | -0.18 | 3.80 | 3.33 | 1.47 |
| *RMSE* | 0.96 | 3.86 | 3.94 | 1.66 |
| **Coastal EA** | | | | |
| Casey | **-0.94** | **-3.5** | **-3.92** | **-3.38** |
| Davis | **-1.28** | **-2.03** | -1.56 | **-1.29** |
| Dumont Durville | **-0.53** | **-3.21** | **-3.56** | **-2.76** |
| Mawson | -0.28 | **-2.62** | **-2.84** | **-2.53** |
| McMurdo | **-3.45** | **-2.33** | **-3.12** | **-3.38** |
| Mirny | **1.57** | -0.32 | -0.01 | 0.08 |
| Novolazarevskaya | 0.38 | **-2.33** | **-1.78** | **-1.33** |
| Scott Base | **-1.36** | 1 | 0.43 | 0.01 |
| Syowa | **-2.31** | -0.53 | -1.43 | -0.75 |
| *Mean error* | -0.91 | -1.76 | -1.98 | -1.70 |
| *RMSE* | 1.65 | 2.26 | 2.44 | 2.13 |
| **Ice shelves** | | | | |
| Halley | **2.68** | **6.84** | **7.54** | **5.38** |
| Neumayer | **3.21** | **5.45** | **6.58** | **5.25** |
| *Mean error* | .95 | 6.15 | 7.06 | 5.32 |
| *RMSE* | 2.96 | 6.18 | 7.08 | 5.32 |
| **Peninsula** | | | | |
| Bellingshausen | **-0.86** | 0.3 | 0.11 | 0.08 |
| Esperanza | **-1.66** | 1.32 | -0.76 | -0.9 |
| Faraday | **-1.79** | **-1.23** | **-2.24** | **-2.12** |
| Marambio | **-2.34** | 1.6 | -1 | -1.62 |
| Marsh | **-0.64** | 0.36 | 0.06 | 0.13 |
| Orcadas | **-0.92** | 0.2 | 0.19 | -0.64 |
| Rothera | **-2** | **-0.99** | **-3.14** | **-2.63** |
| *Mean error* | -1.46 | 0.22 | -0.97 | -1.1 |
| *RMSE* | 1.58 | 1 | 1.54 | 1.48 |
| **Southern Ocean** | | | | |
| Gough | **-1.05** | -0.3 | 0.12 | **-0.71** |
| Macquarie | **-0.47** | 0.09 | **0.39** | -0.25 |
| Marion | **-0.92** | **-0.43** | 0.01 | **-0.46** |
| *Mean error* | -0.81 | -0.21 | 0.17 | -0.47 |
| *RMSE* | 0.85 | 0.31 | 0.24 | 0.51 |



**Table B2.** Error on READER weather station $T_{2m}$ in the ARP-AMIP simulation for the reference period 1981-2010. Errors significant at p=0.05 are presented in **bold**.

| Stations | DJF | MAM | JJA | SON |
|---|---|---|---|---|
| **EAP** | | | | |
| Amundsen Scott | 0.47 | **2.4** | 1.06 | 0.94 |
| Vostok | **-1.46** | **3.21** | **3.22** | **1.89** |
| *Mean error* | -0.50 | 2.81 | 2.14 | 1.42 |
| *RMSE* | 1.08 | 2.83 | 2.40 | 1.49 |
| **Coastal EA** | | | | |
| Casey | **-3.97** | **-5.72** | **-6.88** | **-5.41** |
| Davis | **-1.61** | **-4.19** | **-5.98** | **-3.31** |
| Dumont Durville | -0.45 | **-2.82** | **-4.07** | **-2.24** |
| Mawson | **-2.24** | **-4.32** | **-5.67** | **4.26** |
| McMurdo | **-7.13** | **-6.48** | **-8.11** | **-8.38** |
| Mirny | **-1.24** | **-2.21** | **-2.97** | **-1.98** |
| Novolazarevskaya | **2.49** | 0.58 | -1.02 | 0.58 |
| Scott Base | **-5.03** | **-3.15** | **-4.56** | **-4.98** |
| Syowa | -0.17 | -0.58 | **-1.49** | 0.04 |
| *Mean error* | -2.15 | -3.34 | -4.53 | -3.33 |
| *RMSE* | 3.46 | 3.86 | 5.06 | 4.25 |
| **Ice shelves** | | | | |
| Halley | **1.27** | **2.45** | 1.21 | 0.88 |
| Neumayer | **2.18** | **1.21** | 0.9 | **1.41** |
| *Mean error* | 1.73 | 1.83 | 1.06 | 1.15 |
| *RMSE* | 1.78 | 1.93 | 1.07 | 1.18 |
| *Peninsula* | | | | |
| Bellingshausen | **-1.02** | -0.42 | -0.24 | -0.08 |
| Esperanza | **-1.1** | 0.5 | -1.33 | -0.88 |
| Faraday | **-2.66** | **-4.66** | **-5.74** | **-3.66** |
| Marambio | **-1.87** | 1.04 | -1.27 | -1.6 |
| Marsh | **-0.81** | -0.36 | -0.29 | -0.03 |
| Orcadas | **-1.13** | -0.04 | 0.61 | -0.76 |
| Rothera | **-5.55** | **-7.88** | **-8.72** | **-6.13** |
| *Mean error* | -2.02 | -1.69 | -2.43 | -1.88 |
| *RMSE* | 2.55 | 3.49 | 4.02 | 2.80 |
| **Southern Ocean** | | | | |
| Gough | **-0.98** | -0.34 | 0.02 | **-0.79** |
| Macquarie | **-0.71** | -0.35 | 0.2 | **-0.45** |
| Marion | **-1.15** | **-0.43** | -0.05 | **-0.68** |
| *Mean error* | -0.95 | -0.37 | 0.06 | -0.64 |
| *RMSE* | 0.96 | 0.38 | 0.12 | 0.66 |


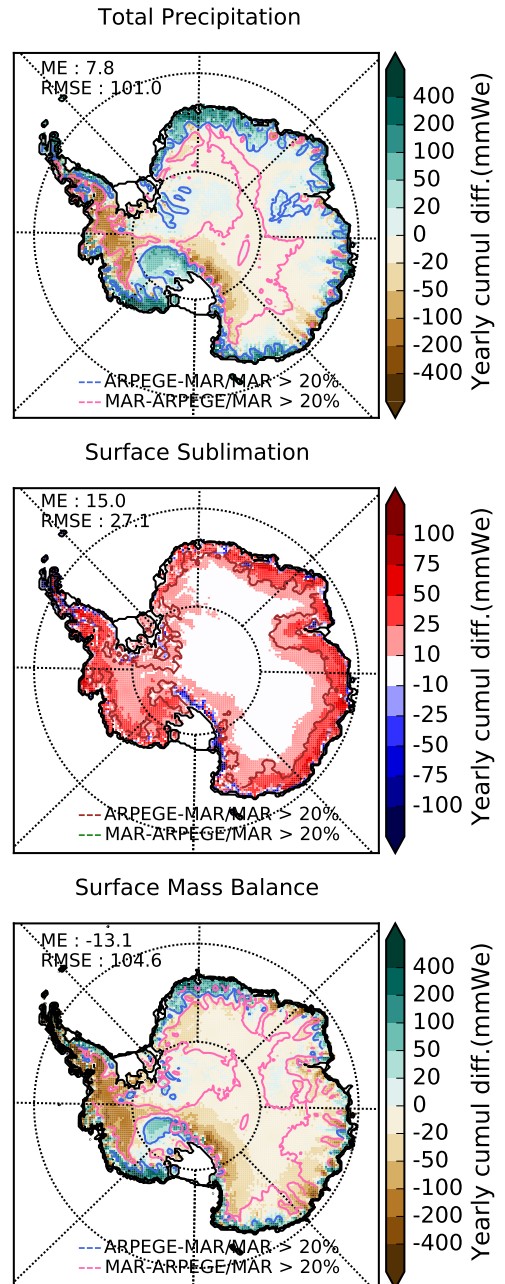

**Figure C1.** Total Precipitations (*top*), surface snow sublimation (*centre*) and SMB (*bottom*) for ARP-AMIP minus MAR-ERA-I yearly cumul difference (mmWe yr$^{-1}$) for the reference period 1981-2010. Pink (brown) and blue (green) contour lines represents areas where ARPEGE-MAR differences are larger than 20%.