# Peer review of "Significant additional Antarctic warming in atmospheric bias-corrected ARPEGE projections with respect to control run"

_The Cryosphere, 2020_

## Referee Comment (RC1) · Anonymous Referee #1 · 19 Nov 2020

General comment:

This study consists of two parts: implementation of simulated bias correction (with respect to a reanalysis dataset) in an atmospheric general circulation model (AGCM) and the effect of bias correction on assessing the future climate change in the Antarctica. The use of the AGCM with refined resolution near the Antarctica allows one to acquire detailed, useful information with only prescribed lower (oceanic) boundary conditions and without lateral (regional) boundary conditions. It can be seen as a type of dynamical downscaling. The authors implemented a method of correcting biases in the model, and their approach is, I think, unique for this context, and potentially very practical. The description of the methodology (Sect. 2) is clearly given and it is easy to follow the text throughout. The authors demonstrated that the result differs significantly with and

without the bias correction. I found that the first part of the study (implementation of the bias correction) (Sects. 3.1 and 4.1) is nicely done but the second part (Sect 4.2) needs significant improvement as argued below.

Major comment:

1. As in the title, abstract, and conclusion, the most important scientific finding in this study is that the climate change signal is assessed differently with and without the bias correction. The heart of the discussion should then be placed on whether the bias-corrected climate change simulations are more reliable or not compared to the uncorrected climate change simulations. It is not trivial because the addition of extra terms to the tendency equations violates the conservation laws of physics and may distort the processes operating in the model. There are at least two ways to check the validity of the approach. The first method is a perfect model study in which the bias in a model with respect to the simulated present-day climate by another model is to be corrected and one can investigate whether the climate change signal simulated by the second model is better reproduced by the bias-corrected first model. The second method is to provide a physically persuasive mechanism/rationale why the bias-corrected simulations are more reliable. Now, the first approach requires two models and too much extra work, and still the result may depend on the reference models used. The second approach is, on the other hand, feasible and essential.

In Sect. 4.2.1, the authors describe how the changes in atmospheric circulation and sea level pressure are different with and without the bias correction, but they do not discuss the reason and mechanism. It is, thus, difficult to assess which result is more reliable. Before discussing the difference from previous studies, they should investigate and explain the mechanism in their own model. In Sect. 4.2.2, it is stated that the difference in summer SAM change has large impact on the surface warming, but again they do not investigate why the summer SAM is different and which dynamical mechanism responsible for it is affected by the imposed bias correction. Moreover, they cite PSA1, PSA2, and Amundsen Sea Low differences as potentially important to

understand the temperature difference with and without the bias correction, but they do not explain the link of these differences to the background climate state. In Sect. 4.2.3, the difference in precipitation change is attributed to the atmospheric circulation difference, but the link between the unperturbed climate state and the circulation difference is not explained. To my view, these are the most important scientific points of the paper, and the necessary data to explore are all at the authors' hands. Without these explanations, one may see the paper as simply demonstrating how the present-day simulation becomes close to the reanalysis dataset after the correcting terms are added (as implemented), and the simulated future climate change signal are affected by the bias correction for unknown reasons. As it is unclear which one (bias-corrected or uncorrected) is more realistic in the climate change simulations, one could argue that the main conclusion is not convincingly established.

2. As the tendency terms are corrected at each time step so that it should reproduce the reanalysis datasets, it is not surprising that the simulated result shows better agreement with the targeted dataset. I would not describe it as "improvement" as the authors claim. It only demonstrates that the implementation worked as designed. I do not at all mean that it is easy to achieve it (indeed I appreciate the effort and see that the approach has a great potential), but this part of the study is more a technical advancement, rather than a scientific one. Please highlight which results are unexpected (or surprising) in simulating the present-day climate with the bias correction and implication of those unexpected results. The climate modelling community had taken the similar approach of so-called flux adjustments decades ago and virtually abandoned it by now due to various side effect. I am a bit surprised that the authors do not touch upon such a historical path in the model development and not discuss why the authors consider it worth to be revived.

Minor comment:

1. In Fig. 1-(b)-left, is there explanation why the SLP bias in the vicinity of Antarctica does not disappear after the bias correction?

---

## Referee Comment (RC2) · Anonymous Referee #2 · 16 Dec 2020

Review of "Significant additional Antarctic warming in atmospheric bias-corrected ARPEGE projections" by Julien Beaumet et al.

With great interest I have read this manuscript which is generally well written. I do not see the need summarize the content of the manuscript.

After reading the manuscript and collecting my comments, I've read the first review. I had also caught my attention that the discussion of tendency-corrected projection is less in depth as the discussion of the historical period. I know, verifying a projection is much more complicated as we do not know the future.

The approach proposed here assumes stationary of the model biases while the atmospheric mean state is changing due to anthropogenic forcings. It would be good if the

authors provide more detail on the added/removed energy, moisture and momentum and how this relates to the total global energy, moisture and momentum budget. If the local/total tendency correction is significant compared to the local/total energy/moisture budget, this reduces the reliability of projections, as a different mean state would likely need a different mean tendency correction.

Other significant comments:

The title needs to be revised as it is not self-evident on with respect to what the additional warming has been observed.

I would propose to merge section 4.1 with 3.1 as 4.1 is an extended evaluation of discussion of the Present Climate, and hence the continuation of 3.1. It would connect 3.2 and 4.2 as well.

The discussion of SMB and precipitation (section 3.1.4) can be more clear, I propose to separate the discussion and figures concerning of precipitation and SMB. I would propose to dedicate section 3.1.4 on precipitation only, discussing the differences between ARP-AMIP-AC, ARP-AMIP, MAR and RACMO as is done now. Given the substantial differences between ARP-AMIP-AC and MAR (and RACMO), I would like to see figures of at least also the modelled precipitation (thus not a difference plot) of ARP-AMIP-AC and (MAR and/or RACMO). Possibly the new Figure 5 can combine these 2/3 precipitation plots with the old figures 5a, 6 and 7a,b. It would be good for the text, for example, as figures 5a and 7b are close/next to each other so that they can be compared easily. It might be worth considering reorganising P13L3 to P13L22 so that it becomes easier to read and grasp. Furthermore, anyone should be aware that MAR and RACMO2 are not the real truth of the precipitation, so in the discussion here the authors could take that into account. Hence, take the assessment of Agosta & van Wessem on the performance of their models against observations across the continent into account when MAR and RACMO2 disagree. In 3.1.5 (or at the end of 3.1.4 if 3.1.5 would become too short), the other SMB components and the SMB is discussed, e.g. Figure 5b,c.

It would be good to connect the dots: too high surface temperatures (-> too high LW emission?), too strong exchange in the ABL, too high sublimation. Of course, if these dots connect in your view. (If not, please argue why so in the reply to the review).

Other textual comments:

P1L16: What's dramatically? It has an emotional load this sentence not necessarily needs. I'll prefer it to be changed into something like considerably.

P1L20: "Fails to compensate" I'll prefer a more neutral expression. Less drama, please.

P2L2: Compared to the "dramatically" at P1L16, this "dramatically" is justified. . .

P2L2: As you are going to argue this SMB increase per K is higher, I would formulate this sentence slightly different as it now reads as a closed case. E.g. ". . ., existing studies align on an increase of the SMB of 5+1 % K-1 (. . ."

P2L5: As models seems to align, which uncertainties are missed? Please expand. Similarly, please rephrase "In this regard" as it may seem to refer back to the SMB - Temp relation, but you do refer to the potential mass loss of the AIS.

P3L27: I would like to see a figure of the grid. A reference "(see <paper>, Fig. <#>)" will do.

P4L3: Please state clear that GELATO is only needed for a correct SEB over sea ice.

P5L7: Brackets is not the most elegant option. ". . .conditions, e.g. greenhouse gas concentrations, and . . ."

P5L32: I can imagine arguments to use ERA-Interim here even though it is now superseded by ERA5 for about a year. Nevertheless, provide these/this argument here; briefly of course.

P8, T2: Give a reference to Eq. (4) for \Delta_r E.

P9, F1: add in the caption that Mean SLP is shown.

P7L7: not the uncorrected ARP-AMIP run but the mean SLP of this run is low biased.

P7L20: Is there a specific process-based reason for the remaining warm bias in winter 200 hPa, near surface pressure bias over Antarctica and surface temperature errors (Sec 3.1.3)?. Is it related to radiative problems (too much LW TOA emission, and hence too strong meridional circulation) or too strong horizontal (stratosphere) and vertical (ABL) mixing? If studied before, a reference and short note will do.

P8L11: If I read P6L6 correctly, these BMUs are derived using ARP-AMIP-AC data too. Is there a reason that typical states of ARP-AMIP-AC are nonetheless missed?

P14 F5: 1) The numerous overlays make the graphs hard to assess 2) There is a typo in the caption ".W.e" 3) I don't see the necessity to clip precipitation (Fig. 5a) to the continent only. Please show ocean values as well (without changing the plotted region). That request involves also figures 6, 7 and 10.

P17 F8: it would be good if the legend would make clear too which lines belong to historical simulations and which to projections. Now it doesn't.

P17L7: a larger displacement than with? (NorESM1-M likely).

P17/18 S3.2.2: This section is rather descriptive. What is driving the regional warming? Reduced LW TOA emission, better meridional exchange, or unmodified global warming. And why is much of the southern ocean not warming? I don't think it's wrong/err, but I'm missing explanations even brief.

P19F9: The panels are really small, please blow them up by 50% at least. Grey lines needs to be searched with zoom...

P21F10: for ARP-MIR-21-AOC, changes are well over 75% for about half of Antarctica. Please adjust the scale so that is "colormap clipping" is largely removed.

P21L4: Section. A is likely Appendix A.

P21L8: I personally don't see the logic in trying strong tendency correction on boundary

layer processes if you haven't tried extending the current tendency correction on the boundary layer. Furthermore, it sounds logical to me that systematic errors in the boundary layer representation induce biases at 850 hPa even if this layer has tendency correction. Given that you now have an isolate region with biases (the ABL), retuning is now much more easy as indirectly induced global feedbacks of retuning can be removed by rerunning the tendency correction procedure on the retuned model.

P22L2: A reference to figure B1 (I presume) is missing. Make more clear in the text that you're comparing against MAR as I missed that on first sight. T2m temperature biases over land and ice sheets make only sense if the model topographies are (near) similar. An 100 m elevation difference gives a 0.5 to 0.8 K temperature difference, so the modelled biases over the smaller ice shelves and continent escarpment could potentially explain part of the biases. Of course, similar biases are observed over the larger ice shelves where topographic errors are unlikely. It would be good to add in the appendix a figure with local differences in the model orography along with Figure B1. If these orographic deviations are negligible, that should be stated clearly in the manuscript. Similarly, consider to add model and station elevations in Table B1.

P22L30: "Large discrepancies . . ." This is a very indirect way of saying that MAR and RACMO2 do not agree, right? Please formulate more direct to take away confusion. Furthermore, nice to hear that MAR and RACMO2 don't agree, but what is the implication for this paper? Does ARP-AMIP align better with RACMO2?

P23L27: The conclusions of this paper are slightly more nuanced than this. RACMO2 ignores horizontal transport of falling precipitation and subsequently misses evaporation of snow advected into dryer or warmer locations. If ARPEGE has also no prognostic precipitation, this error could be shared. However, the induced error by missing precipitation advection decreases for decreasing model resolution.

P23L29: ". . . (15 min). Over the . . ." Add something like Furthermore/Finally to make clear that the following is not directly related to the "first Agosta 19 comments".

S4.2.1&4.2.2: It is good to add references to the figures that are discussed. Now the text is dry and requires the reader to have memorized all the graphs implicitly referenced. And, as discussed above, it is less thorough as the rest of the paper.

Figure 1, 8, 9, A1, A2: The labels and legend text in these figures are too small.

Table B1 and B2: I do not see why you cannot merge these two tables into one table.

---

## Author Comment (AC1) · 15 Mar 2021

"**Significant additional Antarctic warming in atmospheric bias-corrected ARPEGE projections**" by J. Beaumet, M. Déqué, G. Krinner, C. Agosta, A. Alias & V. Favier

Author response to referee's comment.
Anonymous Referee #1

General comment:

This study consists of two parts: implementation of simulated bias correction (with respect to a reanalysis dataset) in an atmospheric general circulation model (AGCM) and the effect of bias correction on assessing the future climate change in the Antarctica. The use of the AGCM with refined resolution near the Antarctica allows one to acquire detailed, useful information with only prescribed lower (oceanic) boundary conditions and without lateral (regional) boundary conditions. It can be seen as a type of dynamical downscaling. The authors implemented a method of correcting biases in the model, and their approach is, I think, unique for this context, and potentially very practical. The description of the methodology (Sect. 2) is clearly given and it is easy to follow the text throughout. The authors demonstrated that the result differs significantly with and C1 without the bias correction. I found that the first part of the study (implementation of the bias correction) (Sects. 3.1 and 4.1) is nicely done but the second part (Sect 4.2) needs significant improvement as argued below.

Authors : We thank the referee for their encouraging review and constructive comments on the manuscript. A point by point response to each comment is given below.

Major comment:

1. As in the title, abstract, and conclusion, the most important scientific finding in this study is that the climate change signal is assessed differently with and without the bias correction. The heart of the discussion should then be placed on whether the bias-corrected climate change simulations are more reliable or not compared to the uncorrected climate change simulations. It is not trivial because the addition of extra terms to the tendency equations violates the conservation laws of physics and may distort the processes operating in the model. There are at least two ways to check the validity of the approach. The first method is a perfect model study in which the bias in a model with respect to the simulated present-day climate by another model is to be corrected and one can investigate whether the climate change signal simulated by the second model is better reproduced by the bias-corrected first model. The second method is to provide a physically persuasive mechanism/rationale why the bias-corrected simulations are more reliable. Now, the

first approach requires two models and too much extra work, and still the result may depend on the reference models used. The second approach is, on the other hand, feasible and essential.

Authors : We agree with the reviewer on this relevant remark. Regarding the first point, we will investigate the typical values of the added/removed energy, moisture and momentum associated with the values of the correction terms. Unfortunately, physical tendencies associated with the radiative scheme or the dynamics in our ARPEGE simulations were not saved in the output, but we will take typical values in inter-model studies from the literature (e.g., Cesana et al., 2019, https://doi.org/10.1175/JCLI-D-17-0136.1). Results of this comparison will be shown in the final answer, and added to the supplementary materials of the paper. In the meantime, we can already inform the reviewer that in the paper by Krinner et al., (2019) using the same method with the atmospheric model LMDZ at lower horizontal resolution, the correction terms associated with the tendency errors were found to be much smaller that the tendency associated with the other processes of the model physics (radiative transfer, dynamics…).

Besides, we want to stress the fact that in AMIP-style experiments as those presented in this study, the fact that the oceans are considered as a surface boundary condition and that it can act as an infinite source or sink of energy already distort the laws of energy conservation in the experiment.

On the second point regarding the possibility to demonstrate the validity of the approach for climate change projections using a perfect model study, this is what has been done in Krinner et al., 2020 (https://doi.org/10.1038/s43247-020-00035-0). In this study, three global atmospheric models, one of them being ARPEGE, emulates one of the two other model corresponding coupled climate model (CGCM) considered as the reference (pseudo-reality). Then future projections were performed using the same corrections term and the "bias-corrected" future projections (RCP8.5 scenario) are compared to the original future projections of the AOGCM used as reference in the beginning. Results show that about 70% of the added-value of the bias-correction remains in this perfect model study for RCP8.5 projections at the end of the 21st century. This study demonstrates the validity of the approach even for climates that are significantly warmer than the one in which the bias-correction terms have been built. These results are consistent with the results from Krinner and Flanner., 2019 (https://doi.org/10.1073/pnas.1807912115 ) who showed that a climate model can be easily and automatically identified by its departure from the ensemble mean in both present historical simulation and late 21st century projection (abrupt 4xCO$_2$ scenario). These results together suggest that climate model biases are mostly stationary within (and even beyond) the range of climate changes projected during this century and allow for the use of empirical bias-correction terms derived in present-day climate using observations or reanalysis as references for future climate projections.

Regarding the last point, some physical explanations on why climate models with more poleward/broader jet structure (that agree better with present-day climate) show less poleward shift

in a warmer climate have been proposed in Bracegirdle et al., 2013 and other previous works. The following can be found in Bracegirdle et al., 2013 : "*These mechanisms, put forward by Barnes and Hartmann [2010] and Simpson et al. [2012], both relate shifts in the jet to tropospheric eddy feedbacks that depend on the time mean jet structure. Barnes and Hartmann [2010] suggest that differences in eddy feedback could originate from differences in wave breaking on the poleward side of the sub-polar jet. According to this mechanism a poleward shift occurs when poleward breaking is suppressed and the resulting wider jet extends to higher latitudes. The implication is that models with jets that already exhibit weak poleward wave breaking (and a wider, more poleward, structure) show weak shifts under global warming, since wave breaking is already suppressed in those models. An alternative mechanism relating to eddy feedbacks on the equatorward side of the jet was suggested by Simpson et al. [2012], who show that the tropospheric eddy feedback (and poleward shift) is stronger when the distance between the sub-polar eddy-driven jet and the sub-tropical critical line is smaller. Higher latitude jets with a larger distance exhibit a weaker poleward jet shift due to a weaker latitudinal coherence of eddy momentum flux convergence across the phase speed spectrum. Their results are for a specific case of tropical stratospheric heating in a simplified GCM (sGCM), but may be relevant more generally.*"

We propose to briefly investigate in our corrected and non corrected simulation the upper-level meridional temperature gradient and/or the distance between the sub-polar jet and the sub-tropical critical line such as done in Bracegirdle et al., 2013 for the CMIP5 ensemble. If relevant, the results of these analysis will be added to the discussion and/or the supplementary material. These processes are widely influenced by the interdecadal climate variability and no robust conclusion can be drawn from only two pairs of 30-years simulations. However, when we put them in the context of previous CMIP5 large ensemble analyses (Bracegirdle et al. 2013), we confirm that these processes are likely at play in our simulations (Fig R1).

Besides, the findings Barnes and Hartmann, 2012 (https://doi.org/10.1029/2012JD017469) suggest that the reduced poleward shift of the eddy-driven jet and the lesser deepening of circum-antarctic low pressure systems found in the empirically bias-corrected simulations is indeed likely to be more realistic. Barnes and Hartmann, (2012) showed that the poleward shift of cyclonic wave breaking reaches a poleward limit around 60°S and that wave breaking on the poleward side of the jet will become less frequent for any further poleward shift of the jet. Their results suggest that there is a theoretical limit to how far South the location of the maximum cyclonic wave breaking can move poleward, most likely controlled by the absolute vorticity gradient and that the observed Southern Hemisphere circulation may already be close to this limit. Therefore, we will refer to this paper in our discussion, as well as the previous arguments in order to highlight more in the discussion the reliability of projected circulation changes in bias-corrected simulation.

[Figure]

Fig R1 : Projected twenty first century surface westerly wind maximum poleward shift as function of the latitudinal position in the historical simulation for the CMIP5 ensemble (red and blue symbols) and for the ARPEGE corrected (squares) and uncorrected (circles) projections presented in this study. Modified from Bracegirdle et al., (2013).

In Sect. 4.2.1, the authors describe how the changes in atmospheric circulation and sea level pressure are different with and without the bias correction, but they do not discuss the reason and mechanism. It is, thus, difficult to assess which result is more reliable. Before discussing the difference from previous studies, they should investigate and explain the mechanism in their own model.

Authors : We think that the arguments brought in the previous question concerning the reference between previous studies that established a link between a poleward position of the eddy-driven mid-latitude jet and a reduced poleward shift (Bracegirdle et al., 2013, Barnes and Hartmann, 2012) together with the stability of the added-value of the bias-correction in future projection realized within the framework of a perfect model test in Krinner et al., (2020) are good arguments in favour of a higher reliability of the projected circulation changes in the bias-corrected projections.

Moreover, we want to stress the fact that for future climate projections not only the projected climate change signal (relative difference with present-day climate) matters but also (maybe even more) the absolute mean state of the climate at the end of the 21st century, especially within the framework of impact assessment studies. In this regard, the biases in the non corrected

simulations being, especially for the atmospheric circulation in the high southern latitude, of the same order of magnitude as the projected changes, its is by construction not possible that the climate mean state at the end of the 21st century is more reliable in uncorrected future projections. Investigating the causes of changes in circulation between corrected and uncorrected projections cannot be done with simple diagnostics and would require complex diagnostics such as done in Barnes and Hartmann, (2012) or run-time diagnostics such as radiative kernels (e.g., Soden et al., 2008, https://doi.org/10.1175/2007JCLI2110.1) that are beyond the scope of this study.

In Sect. 4.2.2, it is stated that the difference in summer SAM change has large impact on the surface warming, but again they do not investigate why the summer SAM is different and which dynamical mechanism responsible for it is affected by the imposed bias correction. Moreover, they cite PSA1, PSA2, and Amundsen Sea Low differences as potentially important to understand the temperature difference with and without the bias correction, but they do not explain the link of these differences to the background climate state.

Authors : The projected evolution towards a more recurrent positive phase of the SAM at the end of the current century results from the projected increase in meridional pressure gradient (deepening lows and increasing mid-latitude high pressure systems) associated with the poleward shift of these structures and of the westerly jet caused by the increase in greenhouse gas concentration. Here again, this poleward shift is likely constrained by the absolute vorticity gradient such as suggested by the results of Barnes and Hartmann (2012) and models that are strongly underestimating the meridional pressure gradient and show an equatorward bias in the position of the westerly jet such in the historical climate as in the uncorrected simulation with ARPEGE are prone to overestimate this increase towards a positive phase of the SAM.

The variability of the Southern Annular Mode was found to have larger influence on the temperature anomalies over the East Antarctic Plateau (Marshall, 2007) and over the Antarctic Peninsula (Clem et al., 2016) in summer and autumn.

Therefore, we will modify the text to explain more explicitly why the bias-corrected simulations evolve towards a less pronounced positive phase anomaly in future projections, why it is likely to be more realistic than the uncorrected projections and why this has a large impact on summer temperatures over East Antarctica.

We agree with the reviewer that introducing the PSA1 and PSA2 mode of variability as possible explanations for the differences in climate change between corrected and uncorrected projection without properly introducing these mode of variability and their link to background climate, and without investigating their representation in our climate simulation is scientifically questionable. We think that the impact of these processes are most likely of second order of importance compared to the impact of the correction on the main pattern of the atmospheric general circulation but these processes would nevertheless deserve to be investigated in a separate study. Therefore, we will

delete any reference to the PSA1 and PSA2 mode of variability to interpret our results. The impact of the correction on the position and depth of the Amundsen Sea Low and its projected displacement in future projection being more straightforward and easier to interpret, it will still be discussed but we will introduce more details and reference to previous work in this part of the discussion.

In Sect. 4.2.3, the difference in precipitation change is attributed to the atmospheric circulation difference, but the link between the unperturbed climate state and the circulation difference is not explained. To my view, these are the most important scientific points of the paper, and the necessary data to explore are all at the authors' hands. Without these explanations, one may see the paper as simply demonstrating how the present-day simulation becomes close to the reanalysis dataset after the correcting terms are added (as implemented), and the simulated future climate change signal are affected by the bias correction for unknown reasons. As it is unclear which one (bias-corrected or uncorrected) is more realistic in the climate change simulations, one could argue that the main conclusion is not convincingly established.

Authors : Regarding the reliability of projected precipitation change, we think a good example can be given in the western West Antarctica region (Maria Byrd Land). In this region, the position and depth of the Amundsen Sea Low, currently located at the fringe of the Amundsen and Ross Sea in winter and spring, has a large influence on the advection of moist air and therefore precipitation on these coastal regions. In the uncorrected historical simulation, the position and depth of this climatological pressure minimum is widely biased. Unsurprisingly, total precipitation in this region is better simulated (better agreement with MAR and RACMO and also with CloudSat snowfall, see response to the second reviewer below) in the bias-corrected simulation. Similarly to the example given by Maraun et al., 2017 (https://doi.org/10.1038/nclimate3418) for projected changes in precipitation across western Europe, since the position and depth of this main low pressure system is largely biased in the uncorrected historical simulation, the absolute deepening and displacement of the centre of the climatological low cannot be correct in the uncorrected future projection and so will be its impact on regional precipitation change in coastal areas.

2. As the tendency terms are corrected at each time step so that it should reproduce the reanalysis datasets, it is not surprising that the simulated result shows better agreement with the targeted dataset. I would not describe it as "improvement" as the authors claim. It only demonstrates that the implementation worked as designed. I do not at all mean that it is easy to achieve it (indeed I appreciate the effort and see that the approach has a great potential), but this part of the study is more a technical advancement, rather than a scientific one. Please highlight which results are unexpected (or surprising) in simulating the present-day climate with the bias correction and implication of those unexpected results. The climate modelling community had taken the similar

approach of so-called flux adjustments decades ago and virtually abandoned it by now due to various side effect. I am a bit surprised that the authors do not touch upon such a historical path in the model development and not discuss why the authors consider it worth to be revived.

Authors : We agree with the reviewer that the term "improvement" has been misused here and we will avoid using it in this context, and use the terms "bias reduction" instead.

We think that obtaining a better representation of the daily variability of the large-scale atmospheric circulation, such as evidenced in the application of self-organizing maps on sea-level pressure fields, was a non expected (by construction, the bias-correction is only expected to improve mean state) and interesting result. Krinner et al. (2020) found similar results for both inter-annual and synoptic scale variability in their application until 2100 using the perfect model test framework and so did Kharin and Scinocca (2012) in their application of the method for seasonal prediction (doi:10.1029/2012GL052815).

We will add some historical perspectives about the flux adjustments method in the climate modelling community in the introduction. Interestingly, we note that Guldberg et al., 2005 in their first application of flux correction with ARPEGE model for seasonal forecasting found improved skills mostly in the Southern Hemisphere. More recently, Dommenget and Rezny (2018) (https://doi.org/10.1002/2017MS000947) argued in a pilot study that a transparent, well documented flux correction (which is what we achieve in this study) is more desirable and cheaper in computational cost than model tuning which is involved in the development of many climate model. Unlike model tuning, flux correction does not introduce artificial errors between the submodels of the CGCMs.

Besides, as long as climate models have biases of the order of magnitude as the projected changes, *a posteriori* statistical bias correction will be applied to climate model output in the framework of climate change impact assessment. However, these methods are not able to correct for errors associated with large biases on atmospheric general circulation, particularly projected changes in precipitation (Maraun et al., 2017). In this regard, empirical bias correction (or "flux" correction) such as implemented in this study have large potential benefits. Even though these methods are not perfect and distort slightly the tendency associated with the model physics, they do so to a reduced extent compared to model parameter tuning which is widely implemented in the development of climate models. Moreover, empirical bias correction or "flux" correction can be easily switched off so that its impact on the model performance and on projected changes are well known. This way, using both bias corrected and uncorrected projections, we can assess remaining uncertainties on projected climate change and emphasize what should be the priorities in climate model development in order to reduce these uncertainties.

We will add the reference of Dommenguet and Rezny (2018) in the introduction and refer to it in the discussion (section 4.3). In this later section, we will insist more clearly on the points mentioned

above, precisely the potential assets of flux correction methods with respect to model parameter tuning in climate model development and climate change assessment.

Minor comment:

1. In Fig. 1-(b)-left, is there explanation why the SLP bias in the vicinity of Antarctica does not disappear after the bias correction?

Authors response: This positive bias in the seas surrounding Antarctica, even though substantially reduced, especially in the Amundsen Sea sector, remains mostly in winter and spring. During these seasons these areas witness the formation of rapidly developing and evolving meso-cyclons (polar lows). It is therefore likely that the model, even in the bias-corrected simulation, fails to fully capture the formation of these polar lows. Here are two possible explanation :

- The characteristic time of formation of these cyclones is much smaller than the characteristic time of other larger scale cyclones. Therefore, the relaxation time of 72h used in the first nudged simulation towards climate reanalyses that is used to derive correction terms might be too wide to retrieve the right values of correction terms that should be applied to correct for the model deficiencies in simulating these phenomena.
- Katabatic winds flowing from the ice-sheet towards the coast play a key role in the formation of these meso-cyclones. Besides, the formation of a very stable, cold boundary layer at the surface of the ice-sheet plays a key role in the formation of the katabatic winds. In this study and in the previous one (Beaumet et al., 2019b), we have seen that the version of ARPEGE used in these studies has some deficiencies in capturing the formation of very stable boundary layer at the surface of the ice-sheet in winter (similarly to many climate models), which likely impacts the capacity of the model to reproduce correctly the katabatic winds regime around Antarctica and latter the formation of meso-cyclones over near-by seas. We remind that variables in the boundary layer (<100 m) are not corrected at all in the bias-corrected simulations.

We will briefly mention these hypotheses in our discussion of the remaining biases in the corrected historical simulation.

---

## Author Comment (AC2) · 15 Mar 2021

With great interest I have read this manuscript which is generally well written. I do not see the need summarize the content of the manuscript. After reading the manuscript and collecting my comments, I've read the first review. I had also caught my attention that the discussion of tendency-corrected projection is less in depth as the discussion of the historical period. I know, verifying a projection is much more complicated as we do not know the future.

The approach proposed here assumes stationary of the model biases while the atmospheric mean state is changing due to anthropogenic forcings. It would be good if the authors provide more detail on the added/removed energy, moisture and momentum and how this relates to the total global energy, moisture and momentum budget. If the local/total tendency correction is significant compared to the local/total energy/moisture budget, this reduces the reliability of projections, as a different mean state would likely need a different mean tendency correction.

Author response : We thank the reviewer for their positive comments and constructive feedback on the manuscript. The first reviewer had similar concerns and we made a detailed answer, so we only briefly summarize our response to this question here in two points :

First, we will investigate the typical values of the added/removed energy, moisture and momentum associated with the typical values of the correction terms. Unfortunately, typical tendencies associated with the radiative scheme or the dynamics in our ARPEGE simulations were not saved in the output, but will take those typical values in inter-model studies from the literature (e.g., Cesana et al., 2019). Results of this comparaison will be shown in the final answer, and possibly added to the supplementary materials of the paper.

Second, we argue that the results from Krinner and Flaner (2019) and Krinner et al., (2020) are two strong arguments in favour of a reasonably preserved validity of the tendency correction terms built in present-day climate until at least the end of the current century even in the case of an abrupt  $4xCO_2$  scenario. The first study showed that the biases of each climate (assessed through its departure from the ensemble mean in future projection) is stationary through time, at least until the end of the current century. Each model can actually be easily identified automatically by its bias pattern (departure from the ensemble mean) in present-day and abrupt  $4xCO_2$  scenario.

The second study showed that a majority of the added-value of the tendency correction is preserved at the end of the 21st century using the perfect model test framework and three different

AGCM, their respective CGMCS and RCP8.5 projection as "pseudo-reality". In their result, not only most of the improvement of the mean state is preserved throughout the 21st century, but so it is for interannual to synoptic time scale variability.

Other significant comments:

The title needs to be revised as it is not self-evident on with respect to what the additional warming has been observed.

Authors : Ok, we add "with respect to control projections" at the end of the title in order to make it more explicit.

I would propose to merge section 4.1 with 3.1 as 4.1 is an extended evaluation of discussion of the Present Climate, and hence the continuation of 3.1. It would connect 3.2 and 4.2 as well.

Authors : We agree with the reviewer, and we will reorganize and merge section 3.1 on the results of representation of present day climate and the discussion of these results (4.1) and do the same for section 3.2 and 4.2 (future climate). This will improve the readability of the paper and allow us to delete some repetition.

The discussion of SMB and precipitation (section 3.1.4) can be more clear, I propose to separate the discussion and figures concerning of precipitation and SMB. I would propose to dedicate section 3.1.4 on precipitation only, discussing the differences between ARP-AMIP-AC, ARP-AMIP, MAR and RACMO as is done now. Given the substantial differences between ARP-AMIP-AC and MAR (and RACMO), I would like to see figures of at least also the modelled precipitation (thus not a difference plot) of ARP-AMIP-AC and (MAR and/or RACMO). Possibly the new Figure 5 can combine these 2/3 precipitation plots with the old figures 5a, 6 and 7a,b. It would be good for the text, for example, as figures 5a and 7b are close/next to each other so that they can be compared easily. It might be worth considering reorganising P13L3 to P13L22 so that it becomes easier to read and grasp. Furthermore, anyone should be aware that MAR and RACMO2 are not the real truth of the precipitation, so in the discussion here the authors could take that into account. Hence, take the assessment of Agosta & van Wessem on the performance of their models against observations across the continent into account when MAR and RACMO2 disagree. In 3.1.5 (or at the end of 3.1.4 if 3.1.5 would become too short), the other SMB components and the SMB is discussed, e.g. Figure 5b,c.

Authors : Once again, we agree with the reviewer and we will separate the presentation of the result between precipitation and SMB, with a focus on precipitation as the improvement in the representation of precipitation distribution over coastal areas is the most interesting result. We will reorganize the figures and associated comments on precipitation, putting the plots for ARP-AMIP-AC, ARP-AMIP, MAR and RACMO and the associated differences in one figure, which will indeed facilitate the reading and comprehension of the paper. We acknowledge that MAR and RACMO2 are not the truth and will take into account the results of the comparison with observation in Agosta & van Wessem in our discussion. Besides, we compared precipitation from ARPEGE historical simulation with those from the CloudSAT climatology (Palerme et al., 2014). Even though this comparison has very limited validity (only 4 years available for CloudSAT), we can see an overall increased agreement in ARP-AMIP-AC with respect to ARP-AMIP in many coastal areas and an overall reduction in RMSE statistic (see Fig.R2 below). We will consider adding these figures to the supplementary materials of the paper.

It would be good to connect the dots: too high surface temperatures (-> too high LW emission?), too strong exchange in the ABL, too high sublimation. Of course, if these dots connect in your view. (If not, please argue why so in the reply to the review).

Authors : We agree with the reviewer and think indeed that ARPEGE warm biases near the surface, together with the inadequacy of its boundary layer parametrization for a correct modelling of very stable boundary layer, causes the model to overestimate exchanges and turbulent mixing in the ABL and therefore surface sublimation. We will connect these dots in the text.

Other textual comments:

P1L16: What's dramatically? It has an emotional load this sentence not necessarily needs. I'll prefer it to be changed into something like considerably. Authors : Ok, we changed "dramatically" for "substantially".

P1L20: "Fails to compensate" I'll prefer a more neutral expression. Less drama, please. Authors : Ok, we modified "fails to compensate" by "is now largely overtaken by the"

P2L2: Compared to the "dramatically" at P1L16, this "dramatically" is justified. . .

P2L2: As you are going to argue this SMB increase per K is higher, I would formulate this sentence slightly different as it now reads as a closed case. E.g. ". . ., existing studies align on an increase of the SMB of 5+1 % K-1 (. . ."

Authors : We rephrase this sentence in the following way : "..., existing studies agree on the fact that it is expected to increase at a rate of 5+-1 % K-1(..."

P2L5: As models seems to align, which uncertainties are missed? Please expand. Similarly, please rephrase "In this regard" as it may seem to refer back to the SMB -Temp relation, but you do refer to the potential mass loss of the AIS.

Authors : We change 'In this regard' for 'therefore' : "Therefore, it is crucial to reduce the uncertainties on Antarctic regional warming and changes in SMB, in order to assess the SMB negative contribution..." and we add the following sentence thereafter : "Main source of uncertainties arise from poorly represented sea surface conditions and changes in atmospheric general circulation over southern high latitudes in most climate models (Turner et al., 2013, Bracegirdle et al., 2013)."

P3L27: I would like to see a figure of the grid. A reference "(see <paper>, Fig. <#>)" will do.

Authors : Two figures representing the grid spacing and the topography over Antarctica for the configuration used in this study are can be found in J. Beaumet Ph.D dissertation

( ttps://tel.archives-ouvertes.fr/tel-02145468). However, for convenience this figure is presented below and we will consider adding it to the supplementary material of the paper.

---

## Author Response (AR1)

**"Significant additional Antarctic warming in atmospheric bias-corrected ARPEGE**

projections" by J. Beaumet, M. Déqué, G. Krinner, C. Agosta, A. Alias & V. Favier

Author response to referee's comment.

Anonymous Referee #1 Received and published: 19 November 2020

**General comment:**

This study consists of two parts: implementation of simulated bias correction (with respect to a reanalysis dataset) in an atmospheric general circulation model (AGCM) and the effect of bias correction on assessing the future climate change in the Antarctica. The use of the AGCM with refined resolution near the Antarctica allows one to acquire detailed, useful information with only prescribed lower (oceanic) boundary conditions and without lateral (regional) boundary conditions. It can be seen as a type of dynamical downscaling. The authors implemented a method of correcting biases in the model, and their approach is, I think, unique for this context, and potentially very practical. The description of the methodology (Sect. 2) is clearly given and it is easy to follow the text throughout. The authors demonstrated that the result differs significantly with and C1 without the bias correction. I found that the first part of the study (implementation of the bias correction) (Sects. 3.1 and 4.1) is nicely done but the second part (Sect 4.2) needs significant improvement as argued below.

**Authors : We thank the referee for their encouraging review and constructive comments on the manuscript. A point by point response to each comment is given below.**

**Major comment:**

1. As in the title, abstract, and conclusion, the most important scientific finding in this study is that the climate change signal is assessed differently with and without the bias correction. The heart of the discussion should then be placed on whether the bias-corrected climate change simulations are more reliable or not compared to the uncorrected climate change simulations. It is not trivial because the addition of extra terms to the tendency equations violates the conservation laws of physics and may distort the processes operating in the model. There are at least two ways to check the validity of the approach. The first method is a perfect model study in which the bias in a model with respect to the simulated present-day climate by another model is to be corrected and one can investigate whether the climate change signal simulated by the second model is better reproduced by the bias-corrected first model. The second method is to provide a physically persuasive mechanism/rationale why the bias-corrected simulations are more reliable. Now, the first approach requires two models and too much extra work, and still the result may depend on the reference models used. The second approach is, on the other hand, feasible and essential.

Authors : We agree with the reviewer on this relevant remark. Regarding the first point, we have investigated the typical values of the added/removed energy and moisture associated with the typical values of the correction terms (see Fig R1). Unfortunately, typical tendencies coming from the model physics (e.g., radiative scheme) or from the dynamics in our ARPEGE simulations were not automatically saved in the output. Instead those typical values in inter-model studies from the literature (e.g., Cesana et al., 2019). Our analysis

suggests that the typical values for the applied correction for temperatures at midtropospheric level (~500 hPa), are of about 0.0015 K/day for median of absolute values and of 0.0037 K/day for maximum of absolute values. This represents only a few percent of tendencies typically associated with the radiative processes in climate modes : for instance typical values of tendencies associated with radiative (short-wave and long-wave) heating rates of 0.5 to 2.5 K/day are found in Cesana et al., 2019. This analysis and Figure R1 have been added to the supplementary material, and a reference to this analysis has been added in the method section of the paper. Additionally, we also inform the reviewer that in the paper by Krinner et al., (2019) using the same method with the atmospheric model LMDZ at lower horizontal resolution, the correction terms associated with the tendency errors were indeed found to be much smaller that the tendency associated with the other processes of the model's physics (radiative transfer, dynamics...).

---

## Author Response (AR2)

**Significant additional Antarctic warming in atmosphericbias-corrected ARPEGE projections with respect to control run**
Beaumet J. et al., 2021

*Authors response to reviewers comment, second iteration, minor revision.*

*Authors : We thank once more the two reviewers and the editor for their constructive criticism about the manuscript and for the review process.*

Reviewer 1 :

Thank you very much for your effort in replying to and incorporating my comment. I found that all my concerns are satisfactorily answered either in the text or in replies. I note, however, that I do not support a specific reply:

"Besides, we want to stress the fact that in AMIP-style experiments as those presented in this study, the fact that the oceans are considered as a surface boundary condition and that it can act as an infinite source or sink of energy already distort the laws of energy conservation in the experiment."

because one violation does not justify another violation of physical laws, and the governing equations for the atmosphere can still be satisfied in AMIP-style experiments with given boundary conditions, i.e., what comes into the atmosphere equals to what comes out of the atmosphere (a very different level of "violation" from having sources or sinks within the atmosphere).

*Authors : We take note of this relevant remark and will avoid using this argument as a justification in future works and discussion.*

As the revised manuscript does not depend on this specific point, and the justification was made by other arguments, I recommend the manuscript be published in the Cryosphere.

Reviewer 2 :

I would like to thank the authors that they have used the time to address the concerns of the reviewers as good as possible. My evaluation is that the paper can be published almost as is. I have only three minor comments:

P18L11: It looks like the authors would like to say that their results are at the middle of the road compared to MIROC and NorESM1, but that's is not properly put down.

*Authors : We think the reviewer probably meant P18L21. Actually, we wanted to stress out the fact that the climate change signal in terms of strengthening and poleward shift of the westerlies is about twice as low in the bias corrected ARPEGE projections with respect to non corrected control projections. We changed this sentence in order to make it hopefully clearer : « Each future projection (bias-corrected or not) displays a strengthening and poleward movement of the westerly maximum, but the magnitude of these climate change signals are about 50% smaller in ARPEGE bias-corrected projections with respect to non-corrected control run. »*

Figure 9: There seems to be a regular pattern in the four absolute precipitation panels. Is that a plotting artefact? If so, try to remove.

*Authors : Indeed, this was a plotting artifact. We removed it as much as possible.*

P23L12: I would prefer to see an introducing sentence introducing the discussion section, listing the points of discussion.

*Authors : Ok, we add the following introducing sentence in the text « In this section, we discuss the realism of projected climate change in bias-corrected projections as well as the future perspectives associated with the method and the results presented in this study.*